# Modality-specific and modality-general representations of subjective value in frontal cortex
Shilpa Dang[1,2], Jessica Emily Antono[1], Igor Kagan [3,4] & Arezoo Pooresmaeili [1,5] ✉

Neuroeconomics theories propose that the value associated with diverse rewards or reward-predicting stimuli is encoded along a common reference scale, irrespective of their sensory properties. However, in a dynamic environment with changing stimulus-reward pairings, the brain must also represent the sensory features of rewarding stimuli. The mechanism by which the brain balances these needs—deriving a common reference scale for valuation while maintaining sensitivity to sensory contexts—remains unclear. To investigate this, we conducted an fMRI study with human participants engaged in a dynamic foraging task, which required integrating the reward history of auditory or visual choice options and updating the subjective value for each sensory modality. Univariate fMRI analysis revealed modality-specific value representations in the orbitofrontal cortex (OFC) and modality-general value representations in the ventromedial prefrontal cortex (vmPFC), confirmed by an exploratory multivariate pattern classification approach. Crucially, modality-specific value representations were absent when the task involved instruction-based rather than value-based choices. Effective connectivity analysis showed that modality-specific value representations emerged from selective bidirectional interactions across the auditory and visual sensory cortices, the corresponding OFC clusters, and the vmPFC. These results illustrate how the brain enables a valuation process that is sensitive to the sensory context of rewarding stimuli.

When we are presented with options for making a choice, our decisions are guided by our expectations of the rewards associated with each option. Theoretical frameworks of value-based decision-making suggest that our brain associates a subjective value to each available option based on their expected rewards (i.e., a valuation process), then compares these values, and makes a final choice[1–6]. In a multimodal dynamic environment, choice options can have fundamentally distinct sensory features and their associated values can change in time. For instance, the sound of a coffee machine, the smell of fresh bread, and the sight of a bottle of our favourite smoothie in the fridge could evoke the pleasant expectation of a nice breakfast and may therefore have similar value for us as we wake up in the morning. But after satiation we may not enjoy the smell of bread or the sight of smoothie as much as we may still be pleased by the sound of coffee machine.

To address the choice problem in a complex environment, as illustrated above, the brain's valuation process should adhere to two fundamental principles. Firstly, it is important to be able to compare and choose between

distinct types of outcomes, hence value representations independent of the specific type of rewards (e.g., juice, money, or social reward) and the stimuli predicting these rewards (e.g., sound of coffee machine or a picture of a smoothie bottle) should exist in the brain[7–12]. This concept is often referred to as "common currency" coding of value[13–17]. Secondly, the encoding of recent stimulus-value associations should include specific information about the sensory features of each available option[18]. These stimulus-specific (also referred to as identity-specific) representations of value enable flexible adaptation to rapid changes in internal states or external stimulus-value associations[19–23], which is crucial in a dynamic environment. Given that generalization across sensory features may conflict with the need for specific representations for each sensory context, the question arises as to how the brain reconciles these conflicting requirements while computing the subjective value. Here, we investigate this question in a dynamic choice environment in which reward-predicting stimuli vary both in their sensory modality (auditory or visual) as well as their reward association history.

[1]Perception and Cognition Lab, European Neuroscience Institute Goettingen - A Joint Initiative of the University Medical Center Goettingen and the Max-Planck-Society, Goettingen, Germany. [2]School of Artificial Intelligence & Data Science, Indian Institute of Technology Jodhpur, Jodhpur, India. [3]Decision and Awareness Group, Cognitive Neuroscience Laboratory, German Primate Center - Leibniz Institute for Primate Research, Goettingen, Germany. [4]Leibniz Science Campus Primate Cognition, Goettingen, Germany. [5]School of Psychology, University of Southampton, Southampton, UK. ✉e-mail: a.pooresmaeili@soton.ac.uk

Orbitofrontal frontal cortex (OFC) and ventromedial prefrontal cortex (vmPFC) are key brain areas involved in the computation of subjective value and guidance of value-based choices[1,17,24–29]. Different lines of evidence have pointed to the potential role of OFC, in particular the lateral OFC, in stimulus-specific valuation[19,22,30–32], whereas vmPFC has been shown to underlie common currency coding of reward value[7–9,11,32–37]. However, most of these insights have been derived from studies focused on visual stimuli and it remains uncertain whether they extend to the representation of the value of reward-predicting stimuli from different sensory modalities. More importantly, the neural mechanisms that generate stimulus-specific representations of value and coordinate these with common currency valuation have remained underexplored. These gaps have been noticed and addressed by a few recent studies[38–40]. Specifically, Shuster and Levy[40] introduced an elegant way of investigating how the representation of subjective value interacts with the sensory features of stimuli through employing reward-predicting options from two different sensory modalities. Stimuli from different sensory modalities have distinct representations at the input level —for instance in the early visual and auditory cortices. This feature provides a unique opportunity to selectively trace the neural mechanisms that generate the representations of value dependent or independent of the sensory context across the brain, from the early sensory areas to the frontal valuation regions. Taking this approach, Shuster and Levy[40] demonstrated the existence of common representations of stimulus value in the vmPFC, whereas modality-specific representations were only observed in the sensory areas.

In the current study, we aimed to build upon these observations[40] and find whether modality-specific stimulus value representations (SVR) also exist in the key valuation regions in the frontal cortex. We were specifically interested in the valuation process in a dynamic foraging situation when trial-by-trial updating of computed values based on tracking modality-specific reward history is necessary. We further sought to understand *how* the putative modality-specific representation in the frontal cortex are generated through using an effective connectivity analysis approach. Lateral and posterior regions of the orbitofrontal cortex receive highly specific and non-overlapping sensory afferent inputs from auditory and visual sensory areas[41–45]. More medial prefrontal areas including vmPFC on the other hand receive few direct sensory inputs[42]. Moreover, past research has shown that reward value modulates early sensory processing[46–50]. Based on these findings, we hypothesized that the representation of each option's value should exist in OFC in a modality-specific manner and in vmPFC in a modality-general manner[40], and that the co-existence of these coding schemes in the frontal valuation regions is enabled through long-range interactions with the respective sensory cortices of each modality.

In order to test these hypotheses, we acquired fMRI data in a value-based decision-making task with a dynamic foraging paradigm adopted from a previous study[50]. In this task, participants were presented with two stimulus options and chose the one they believed was associated with monetary value based on the trial-by-trial history of reward feedbacks. The two options were rewarded in an independent and randomized fashion to simulate foraging behaviour in a varying environment. To test the influence of sensory modality through which reward information was presented, the task was performed under three different conditions: auditory, visual, and audio-visual, where the choice was made either intra-modally (between options from the same sensory domain) or inter-modally (between options form different sensory domains). To test the hypothesis that modality-specific representations in frontal areas were due to a specificity in value processing requirements and not due to the difference in sensory processing requirements of the auditory and visual domains, a control task was also employed. The control task was designed in a way that the sensory processing requirements were identical to the value task, but selection was based on passively following an instruction as to which stimulus to choose, and not on the assessment of options' reward history. Univariate fMRI analyses revealed modality-specific and modality-general value representations in lateral-posterior OFC and vmPFC, respectively. These results were corroborated by an exploratory multivariate analysis, conducted to test the robustness of the univariate results. Effective connectivity analysis of a

network consisting of regions exhibiting value modulations in auditory and visual sensory cortices, lateral and posterior OFC, and vmPFC, revealed how the two types of value representations emerge and guide value-based decisions.

## Results

### Behavioural results

We examined participants' performance in two behavioural tasks (Fig. 1), referred to as the value-based choice (value) and the instruction-based (control) tasks. In both tasks, participants aimed to maximize their performance while making a choice between two stimulus options. Specifically, in the value task, they sought to maximize their reward by choosing the stimulus option that they believed was associated with reward based on the feedback history (for details see Methods). In the control task, participants' objective was to enhance the accuracy of following instructions regarding which option should be chosen. The presented choice options were either from the same modality (two auditory stimuli or two visual stimuli or they were from different modalities (one auditory and one visual stimulus, as shown in Fig. 1A, B). A choice was made between two stimulus options from two sets: S1 or S2. In different blocks, set S1 included intra-modal auditory low pitch, intra-modal visual green, and inter-modal auditory (high or low pitch). Set S2 included intra-modal auditory high pitch, intra-modal visual red, and inter-modal visual (red or green). We will refer to these sets as S1 = {low pitch, green, auditory} and S2 = {high pitch, red, visual}.

In the value task, participants experienced an unpredictable outcome scenario with a dynamic reward structure. In brief, rewards were assigned to the options from different sensory modalities independently and stochastically at random intervals using a Poisson process[51]. On average, a reward was available for delivery on 33% of the trials and they were distributed between the two stimuli options in different reward ratios of {1:3, 1:1, 3:1}. To ensure a balanced choice between both options, two constraints on reward delivery were implemented, known as "baiting" and "change over delay" (COD), which were adopted from previous studies[50,51]. In baiting, a reward assigned to an option remained available until that option was chosen, preventing "extreme exploitation". Additionally, if the participant switched their choice from one option to another, the earned reward feedback was delayed by one trial and delivered only if the participant chose the same option again. This cost, referred to as COD, was used to discourage an "extreme exploration" strategy, where the participant could consume all rewards without any learning by rapidly alternating choices between options.

Overall, the choice pattern in the value task exhibited matching behaviour nearly in accordance with the Herrnstein's Matching Law[52], which relates the choice behaviour to reward ratios {1:3, 1:1, 3:1}, as shown in Fig. 2A for all modality conditions (auditory, visual, audio-visual). Specifically, the choice ratios, which indicate the number of choices made towards one reward option ($S_1$) over another ($S_2$), increased as the $S_1 : S_2$ reward ratio increased. Importantly, the choice patterns were consistent across sensory modalities. This effect was captured by a strong main effect of reward ratio on choice ratio ($F[2,38] = 183.8$, $p < 0.001$) and no significant interaction between reward ratios and options' sensory modality ($F[4,171] = 0.95$, $p = 0.34$) in a two-way repeated-measures ANOVA. Only a relatively weak effect of modality on choice ratios was observed ($F[2,38] = 5.95$, $p = 0.024$), which corresponded to a tendency of participants to choose visual options more often than auditory options in the audio-visual block (see Table S1). Therefore, we collapsed the choice ratios across modalities for a concise presentation of behavioural results (as shown in Fig. 2B). Analysis of reaction times (RT) revealed no significant effect except for slower RTs in audio-visual compared to both auditory and visual conditions, reflecting that choosing between options from different sensory modalities is more difficult than choices made between items from the same modality (for details see the Supplementary Information and Fig. S1).

We next tested whether participants' choices in the value task followed the predictions of our computational framework. The linear-nonlinear-

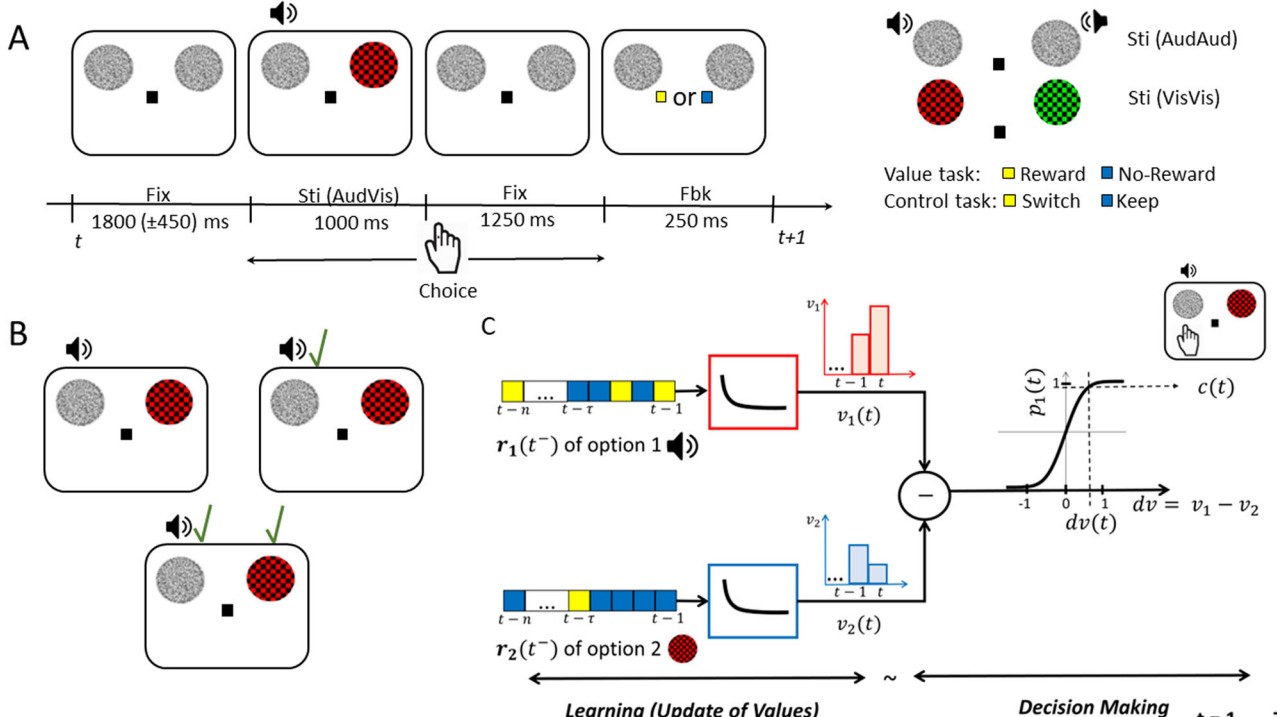

**Fig. 1 | Experimental paradigm and computational framework of choice behaviour. A** General schematic of an audio-visual (AudVis) trial in both behavioural tasks (i.e., value and control tasks). After a jittered inter-trial interval of fixation (Fix), stimuli (Sti) options were presented. Participants made a choice during a response window of 2.25 s from the onset of stimuli, after which the fixation changed to either yellow or blue colour to indicate the feedback (Fbk). In all intervals, two white noise placeholders were shown on the screen. If a choice option was assigned to be from the visual modality, the corresponding placeholder was replaced with a coloured checkerboard (as shown here by a red checkerboard). On the right side of **A** the stimuli options presented during an auditory (AudAud) or a visual (VisVis) trial are displayed. In the value task, the yellow feedback indicated a reward, and the blue feedback indicated no reward, whereas in the control task, the yellow and blue feedback instructed participants either to switch or to keep the past trial's choice, respectively. Please note that auditory tones were presented through MR-compatible headphones. To indicate the side from which each tone was played, we depict a loudspeaker at the corresponding side to where a tone was delivered through the headphones. **B** To create a dynamic multimodal environment for participants in the value task, rewards were assigned to the stimuli options independently and stochastically at random intervals, which means that in a trial there might be a reward for none, either, or both stimuli options, as indicated by green check marks above the rewarded option in three example trials. On average, a reward was available for delivery on 33% of the trials (of a block of the value task), distributed between the options according to reward ratios that were either 3:1, 1:1 or 1:3. Similar to the reward assignment in the value task, switches were assigned to the stimuli options in the control task, such that 33% of trials contained switches distributed according to the ratio 3:1,1:1,or 1:3. **C** Reward history of option 1 and 2; i.e., $r_1(t^-)$ and $r_2(t^-)$; enter as inputs to two identical exponentially decaying filters that weigh rewards based on their time in the past and compute the subjective value of each option (i.e., $v_1$ and $v_2$). The difference of the output of filters gives the differential value between the options (i.e., $dv$). The differential value determines the probability of choice between options (option 1 or 2, here option 1 is chosen as an example) according to a sigmoidal decision criterion.

probabilistic (LNP) model that we used[51] estimated participants' subjective value beliefs for each reward option on a trial-by-trial basis based on the past reward history (see Fig. 1C for a schematic illustration of the model and Methods for details). For this, we approximated the linear filter weights capturing the effect of reward history on choices (quantified by time scale parameter $\tau$; Fig. 2C), and we modeled the probability of choice function (quantified by biasness $\mu$, and sensitivity to value differences $\sigma$; Fig. 2D) for each participant. Across participants, the mean time scale parameter $\tau$ was 1.22 ($\pm 0.15$ s.e.m.), which was significantly greater than zero $t[19] = 8.39$, $p < 0.05$, indicating that choices were in fact most impacted by recent rewards rather than distant rewards in the past (Fig. 2E). Mean biasness $\mu$ across participants was 0.07 ($\pm 0.07$ s.e.m.), which was not significantly different than zero $t[19] = 0.66$, $p = 0.52$, indicating that participants did not have a bias towards any particular option. Finally, the mean sensitivity $\sigma$ across participants was 0.81 ($\pm 0.10$ s.e.m.), which was significantly greater than zero $t[19] = 8.81$, $p < 0.05$ and insignificantly lesser than one $t[19] = 1.95$, $p = 0.07$, indicating that participants were aware of the value difference between options and had indeed adopted an optimal balance between exploration and exploitation ($\sigma = 0$ and $\sigma \gg 1$ for extreme exploitative and explorative tendencies, respectively). Following this optimal strategy, participants were able to harvest 94.94% ($\pm 0.84\%$ s.e.m.) of the total rewards available. The aforementioned fit parameters were first derived from the

model fits to the data of each sensory modality separately and then averaged across modalities. We obtained similar results when each sensory modality was inspected separately (see the Supplementary Information, Fig. S2 and Table S2). These results demonstrate that participants' choice behaviours in the value task were strongly predicted by the filter weights, estimated subjective values, and sigmoidal decision function of the LNP model in all conditions.

In the control task, participants followed the instruction provided by the feedbacks with a high accuracy (i.e., 95.2% ± 1.33% collapsed across keep/switch feedbacks), which indicated that they were aware of the task strategy. Because choices in the control task were explicitly instructed on a trial-by-trial basis, participants' decisions were not expected to reflect past reward history, unlike in the value task. This distinction was effectively captured by the LNP (Fig. 2F): the model's estimates of how much participants updated their beliefs about options' differential value (absDVs) based on the feedback history showed a significant interaction ($F[1,19] = 254.7$, $p < 0.001$) between the type of the task (value or control) and the type of feedback received in the past trial (blue or yellow), see Fig. 2F. Whereas in the value task, absDVs derived from the model showed a significant difference between the two types of feedbacks (mean±s.e.m. = 0.12 ± 0.01 and 0.74 ± 0.03 for blue and yellow feedbacks, respectively, $p < 10^{-11}$), these estimates were not significantly different in the control task (mean±s.e.m. =

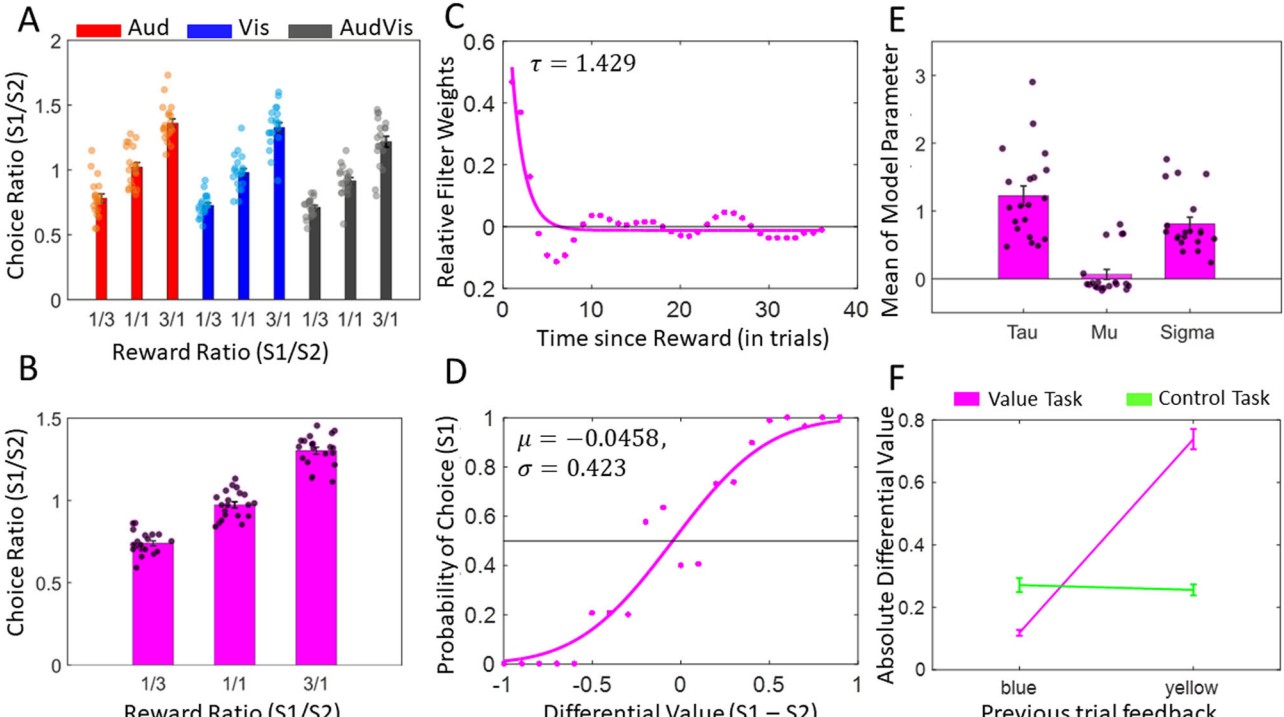

**Fig. 2 | Behavioural results. A** Mean choice ratio across participants for each reward ratio {1:3, 1:1, 3:1} of options $S_1 : S_2$ separately for each modality condition of the value task (*AudAud*, *VisVis* and *AudVis*). **B** Mean $S_1 : S_2$ choice ratios in (A) collapsed across modalities for individual reward ratios. **C** Linear filter weights (dots) and exponential approximation (solid line) showing how past rewards are weighed based on their time in the past for a single participant in the value task. Parameter $\tau$ shows the timescale component of the best-fitting exponentially decaying function. **D** Mapping of differential value of option 1 and 2 to the probability of choice for option 1 (dots) and sigmoidal approximation (solid line) for the same participant as in **C**. Parameters $\mu$ and $\sigma$ of the best-fitting cumulative normal function show the participants' bias towards an option and sensitivity to value differences, respectively. **E** Mean parameters of the best-fitting curves across participants. **F** Relationship between feedback colours (yellow: reward/switch, blue – no reward/keep) and absolute differential value for the two tasks (value and control), across all participants. $S_1 : S_2 = \{lowpitch : highpitch, green : red, auditory : visual\}, S_1 - S_2 = \{lowpitch - highpitch, green - red, auditory - visual\}$. In **A**, **B** and **E** each dot represents the data of one participant. Error bars indicate standard error of the mean (s.e.m.) across participants.

$0.27 \pm 0.02$ and $0.26 \pm 0.02$ for blue and yellow feedbacks, respectively, $p = 0.11$). This finding illustrates that the LNP model effectively captured the distinct roles of feedback in the value and control tasks.

We next examined the robustness of the model in predicting choices in the two tasks, using a leave-one-out procedure described in a previous study [50]. For this, we tested parameter $\tau$ (Fig. 2C, see also Eq. 3) and the sigmoidal choice probability function (solid magenta line as shown in Fig. 2D, and Eq. 7). This leave-one-out procedure was iterated for all 18 blocks of the value task and the mean prediction accuracy across participants was 74.31% ($\pm 1.17$ s.e.m.), being significantly higher than the chance level (paired-sample t-test results when compared with chance level accuracy (50%): t[19] = 20.69, $p < 0.001$). In contrast, when the fit parameters were estimated from the value task and tested on the control task, participants' choices were only predicted at chance level 49.12% ($\pm 1.14$ s.e.m.) (t[19] = -0.77, $p = 0.4504$). These results support our notion that the LNP framework uniquely predicted learning and choices in the value task but not in the control task.

Collectively, our behavioural results confirmed that in the value task participants learned and updated their beliefs about options' value from both sensory modalities through monitoring the feedbacks received on each trial, whereas in the control task they passively followed the instructions without any further processing of stimulus value, as intended.

### fMRI results
#### Modality-general and modality-specific stimulus value representations (univariate): vmPFC and OFC.
We next specified a general linear model (GLM) for each participant to characterise the stimulus value representations (SVRs) in different experimental conditions in the brain and their potential dependence on the sensory modality of stimuli.

Importantly, this GLM included several parametric regressors, referred to as the value regressors, that modelled the trial-by-trial modulations in participants' beliefs regarding the value of each stimulus option as estimated from the computational modelling of the behavioural data. The contrasts that were defined based on these regressors hence assessed the extent to which fMRI responses changed as a function of modulations in subjective value of a stimulus (see Methods, Fig. 1C and Fig. S4, Table 1 and Table S5). The stimulus value representations (SVRs) of each modality condition were then identified in the frontal cortex (see Methods and Fig. S3 for specifications of the search area) based on a group-level random-effects analysis on the contrast images obtained from all participants.

In intra-modal conditions (*AudAud* and *VisVis*), we examined the parametric value regressors separately for the auditory or visual sensory modality (referred to as *intraAudSV > 0* and *intraVisSV > 0* contrasts, respectively). These contrasts reflect the combined responses to both parametric value regressors for each trial within the specified condition. For example, in the auditory condition (*AudAud*), the *intraAudSV > 0* contrast assesses the responses to high- and low-pitch auditory options relative to the baseline (see Methods and Table 1). The auditory contrast in the value task revealed significant activations in vmPFC and left lateral OFC (latOFC) and the visual contrast revealed activations in vmPFC and left posterior OFC (postOFC, Fig. 3A-E). We did not find any significant activation in the right OFC for either of these contrasts. Lateralization of reward responsiveness in OFC could be related to a functional specialization of the left and right lateral OFC and has been reported in the past[53]. Importantly, we found a segregation of value-processing clusters across the two sensory domains in OFC ($d = 20.59$ mm, $d$: Euclidean distance; criterion to decide a segregation was $d > 8$ mm according to the

definitions in[54,55]). On the contrary, the auditory and visual clusters in vmPFC were substantially overlapping (Fig. 3A, B, D).

The aforementioned results hinted towards a functional segregation in the OFC, but not vmPFC, with regard to the representations of stimulus value from the visual and auditory sensory modalities. If this is the case, we

### Table 1 | Stimulus value representations in vmPFC and OFC for various univariate contrasts

| Contrast | Region | *X* | *Y* | *Z* | *t*(19) | *k* |
|---|---|---|---|---|---|---|
| *intraAudSV* > 0 | vmPFC_L | −6 | 64 | −2 | 5.33 | 121 |
| | latOFC_L | −48 | 32 | −10 | 5.31 | 134 |
| *intraVisSV* > 0 | vmPFC_L | −8 | 58 | −14 | 4.70 | 142 |
| | postOFC_L | −30 | 26 | −18 | 6.20 | 143 |
| *interAudVisSV* > 0 | vmPFC | 0 | 52 | −10 | 6.93 | 537 |
| | OFC_L | −36 | 36 | −14 | 7.60 | 242 |

MNI coordinates (x, y, z) and T value corresponds to the local maxima peak of the cluster activations at SVFWE corrected *P* < 0.005 (cluster labels are from AAL atlas[88]). Statistical maps were assessed for cluster-wise significance using a cluster-defining threshold of *t*(19) = 3.58, *P* = 0.001; and using small volume corrected threshold of *P* < 0.005 (referred to as a small volume family-wise-error (SVFWE) correction) within the frontal search volume.

*Description of contrasts*: **IntraAudSV > 0**: contrast capturing responses elicited by changes in subjective value (SV) when choice options consisted of two auditory stimuli (*AudAud*). The contrast corresponds to the overall responses elicited by either low-pitch (*lpSV*) or high-pitch (*hpSV*) auditory stimuli. Thus, this contrast reflects the combined response to both types of auditory stimuli.
**IntraVisSV > 0**: contrast capturing responses elicited by changes in subjective value (SV) when choice options consisted of two visual stimuli (*VisVis*). The contrast corresponds to the overall responses elicited by either green (*gSV*) or red (*rSV*) visual stimuli. Thus, this contrast reflects the combined response to both types of auditory stimuli.
**InterAudVisSV > 0**: contrast capturing responses elicited by changes in subjective value (SV) when choice options consisted of one auditory and one visual stimulus (*AudVis*). The contrast corresponds to the overall responses elicited by either auditory (*aSV*) or visual (*vSV*) stimuli. Thus, this contrast reflects the combined response to both types of stimuli.
*see also Table S5 for contrasts defined based on each parametric value regressor.

should observe both types of representations (i.e., SVRs for visual and auditory modality) in the inter-modal condition where the values of both sensory modalities are simultaneously tracked. This is exactly what we observed when we examined the modulation of fMRI responses by the value of the two options in the inter-modal condition (*AudVis*). This analysis (referred to as *interAudVisSV* > 0 contrast in Table 1) showed significantly activated clusters in vmPFC and in the left OFC (Fig. 3F). Interestingly, within the cluster in the left OFC (cluster peak at -36, 36, -14, see Table 1), there were two local maxima peaks (-44, 26, -12, t(19) = 5.24 and -36, 24, -18, *t*(19) = 4.61), which were adjacent (d < 8 mm) to the respective auditory and visual peaks found in the intra-modal conditions. While the above results in intra- and inter-modal conditions were based on contrasting both parametric regressors (one for each choice option) against the baseline, we obtained similar results when each parametric regressor was separately contrasted against the baseline (see Supplementary Information and Table S5).

To test whether the observed segregation of auditory and visual SVRs in the OFC during intra-modal conditions truly reflected a functional specialization for modality-specific valuation, we performed a cross-validation procedure (Fig. 3G). For this, the definition of regions of interest (ROIs) and the measurement of effect were done on independent datasets to avoid double-dipping[56]. Specifically, we defined the visual and auditory ROIs based on the corresponding stimulus value representations from the inter-modal condition. We then assessed the responses of these ROIs to individual parametric value regressors for each stimulus option in the intra-modal conditions (i.e., visual red, visual green, auditory high-pitch, and auditory low-pitch) and averaged the responses within each modality (see Methods, Supplementary Information, and Table S5 for details).

Modality-specific valuation would require that the auditory ROI is more responsive to variations in the subjective value (SV) of auditory stimuli compared to visual stimuli, and vice versa for the visual ROI. Statistically,

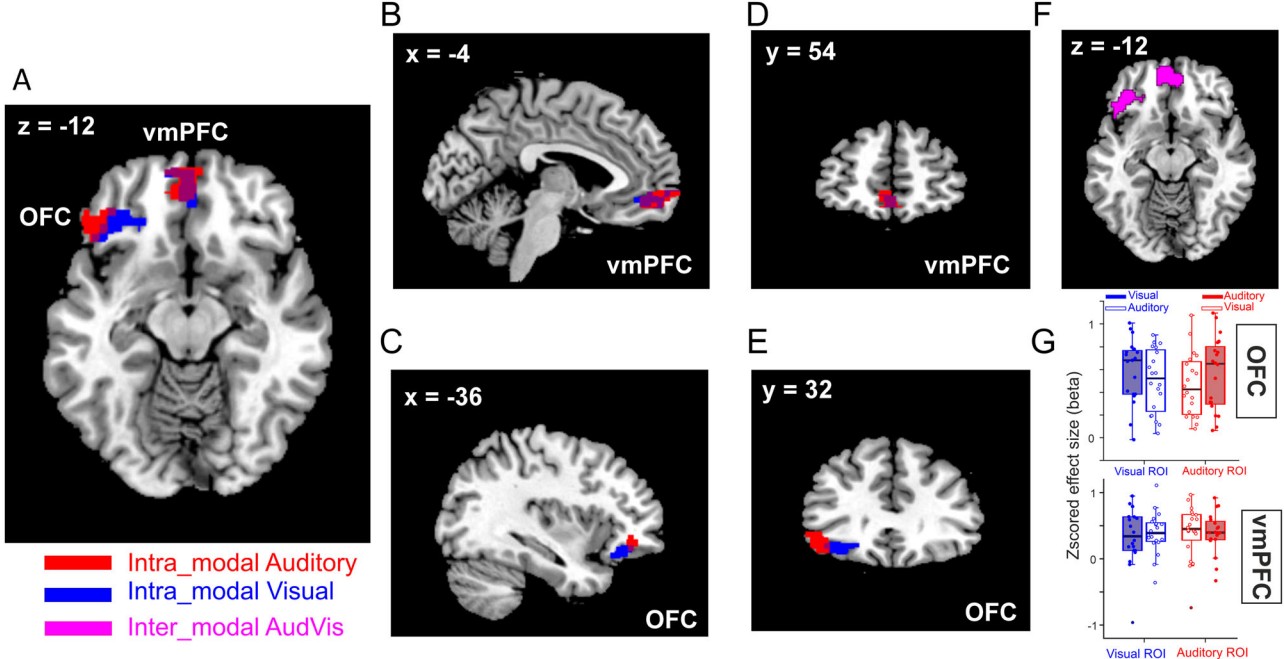

**Fig. 3 | Stimulus value representations (SVR) across different sensory modalities.** **A-E** In intra-modal conditions, we observed segregated modality-specific SVRs in the OFC: for the auditory modality in the left lateral OFC (red cluster) and for the visual modality in the left posterior OFC (blue cluster). Overlapping modality-general SVRs were found in the vmPFC for both auditory and visual modalities. **F** In inter-modal conditions, SVRs were observed in both the OFC and vmPFC. All cluster activations in **A-F** are significant at SVFWE corrected *P* < 0.005. **G** To test the dependence of SVRs in OFC and vmPFC (upper and lower panel, respectively) on sensory modality, visual and auditory ROIs were defined based on the corresponding SVRs in inter-modal condition (blue – visual, red – auditory) and tested on intra-modal condition. Filled box plots show the responses to the stimulus value from the same modality as the ROI and unfilled box plots show the responses to the stimulus value in a different modality.

**Table 2 | Stimulus value representations in OFC for various interaction contrasts**

| Contrast | Region | X | Y | Z | t(19) | k | SVFWE corr |
|---|---|---|---|---|---|---|---|
| *Value_all <sub>Parametric</sub> > Control_all <sub>Parametric</sub>* | **postOFC_L** | −24 | 24 | −12 | 6.29 | 584 | *P* < 0.001 |
| | **postOFC_R** | 30 | 24 | −10 | 7.42 | 524 | *P* = 0.001 |
| *Value_all <sub>Trial identity</sub> > Control_all <sub>Trial identity</sub>* | No surviving clusters | | | | | | |

Statistical maps were assessed for cluster-wise significance using a cluster-defining threshold of *t*(19) = 2.86, *P* = 0.005; and using small volume corrected threshold of *P* < 0.005 (referred to as a small volume family-wise-error (SVFWE) correction) within the frontal search volume. MNI coordinates (x, y, z) and T value corresponds to the local maxima peak of the cluster activations and activations shown in bold survive a SVFWE corrected *P* < 0.005.
**Description of contrasts**: *Value_all <sub>Parametric</sub>*: denotes all parametric regressors in both intra-modal (*AudAud, VisVis*) and inter-modal (*AudVis*) conditions of value task, here *lpSV, hpSV, gSV, rSV, aSV, vSV*.
*Control_all <sub>Parametric</sub>*: denotes all parametric regressors in both intra-modal (*AudAud, VisVis*) and inter-modal (*AudVis*) conditions of control task.
*Value_all <sub>Trial identity</sub>*: denotes the unmodulated regressors that modelled the modality of stimuli in a trial (visual, auditory, or audio-visual) in all conditions of the value task.
*Control_all <sub>Trial identity</sub>*: denotes the unmodulated regressors that modelled the modality of stimuli in a trial (visual, auditory, or audio-visual) in all conditions of the control task.

this would manifest as an interaction effect between ROI and the type of SV that the ROI represents. Conversely, modality-general valuation would require that auditory and visual ROIs respond similarly to both auditory and visual SVs. Examining auditory and visual ROIs in the OFC (Fig. 3G, upper panel) and vmPFC (Fig. 3G, lower panel) provided evidence for these types of valuation, respectively. A repeated measures ANOVA with factors: ROI (visual or auditory) in the OFC, and SV (visual or auditory), showed only an interaction between ROI and SV modality (F[1,19] = 9.88, p = 0.005), with no main effect of ROI or SV (all ps > 0.1). The interaction effect corresponded to numerically stronger responses of the auditory ROI to auditory SVs (mean ± s.e.m: 0.57 ± 0.07) than to visual SVs (0.47 ± 0.01, Cohen's D = 0.28), and similarly stronger responses of the visual ROI to visual SVs (mean ± s.e.m: 0.60 ± 0.06) than to auditory SVs (0.50 ± 0.06, Cohen's D = 0.28). However, in both cases, the differences between responses to the same versus different modalities did not reach statistical significance (ps > 0.1). The same analysis in the vmPFC revealed no significant main or interaction effects (all ps > 0.1). These results further support distinct representations of subjective values for each modality in OFC, whereas in the vmPFC subjective value is represented regardless of the stimulus modality.

An alternative explanation for the segregation in modality-wise representations in the OFC is that rather than reflecting the functional specialization of neural responses for the visual and auditory value processing, they reflected differences in the sensory properties of stimuli. To rule out this possibility, we next examined the control task (see Table 2 and Table S4, Table S6 and Fig. S5). Crucially, the modality-specific activations in OFC were absent in the control task when the same contrasts as in the value task were examined, demonstrating that OFC representations reflect the trial-by-trial updating of stimulus-value associations rather than the sensory features of stimuli or simple choice based on the instruction. On the contrary, in the control task activations overlapping with the modality-general representations in vmPFC were found (Fig. S5). Inspecting contrasts which explicitly tested the value and the control task against each other supported these results (Table 2). Contrasting the unmodulated trial identity regressors which modelled the responses to different sensory modalities revealed no significant difference between the two tasks, confirming that the control task was powerful enough to capture the sensory responses to the stimuli. Contrasting the parametric regressors however showed strong bilateral activations in postOFC in the value task, supporting the involvement of the OFC in representing the value of stimuli beyond their sensory properties. The lack of activations in the vmPFC for this contrast further highlights a general role of this area in representing the final choice irrespective of whether or not choices were informed by value or were instructed.

Together, the findings of our univariate analyses provide evidence that the valuation of stimuli from auditory and visual sensory modalities is confined to segregated loci in the OFC. Additionally, our results indicate that the representation of stimulus value is independent of sensory modality in the vmPFC and that this region is involved in processing information related to the final choice across different tasks.

**Table 3 | Stimulus value representations outside of valuation regions**

| Region | X | Y | Z | t(19) | K |
|---|---|---|---|---|---|
| SFGmed | 0 | 54 | 40 | 10.50 | 64 |
| Caudate_L | −14 | 24 | 8 | 10.37 | 67 |
| **visSen_L** | **−20** | **−90** | **2** | **9.75** | **81** |
| **visSen_R** | **24** | **−96** | **10** | **8.86** | **119** |
| Hippoc_L | −36 | −38 | −12 | 9.62 | 30 |
| Hippoc_R | 34 | −22 | −12 | 9.51 | 29 |
| **audSen_L** | **−66** | **−30** | **−4** | **9.08** | **24** |
| **audSen_R** | **66** | **−12** | **−4** | **8.99** | **15** |
| Angular_L | −42 | −56 | 26 | 8.62 | 55 |

MNI coordinates (x, y, z) and T value corresponds to the local maxima peak of the cluster activations at FWE corrected *P* < 0.05 (cluster labels are from AAL atlas[88]). SFGmed – medial Superior Frontal Gyrus; Hippoc – Hippocampus; visSen – Visual Sensory Cortex; audSen – Auditory Sensory Cortex. Highlighted (in bold) activations were used as ROIs in the effective connectivity analysis.

## Stimulus value representations outside of frontal valuation regions: whole-brain univariate analysis

In order to identify regions exhibiting value modulations outside the valuation regions, specifically in auditory and visual sensory cortices, we performed a whole-brain analysis using the GLM described previously. For this purpose, we estimated the overall effect of the parametric value regressors across all conditions in the value task (*AudAud, VisVis* and *AudVis*), which revealed bilateral activations in the auditory and visual cortices (whole-brain FWE corrected *P* < 0.05, cluster size *k* > 10 voxels; Table 3, Fig. S6A-B). When estimating the value modulations for individual conditions separately (auditory, visual), we found modality-specific activations in respective sensory cortices only (see Fig. S6C-D), whereas in inter-modal condition both sensory cortices were activated (see Fig. S6E-F).

In addition to sensory cortices, we found significant value modulations in areas involved in processing different aspects of value-related information, such as detecting the reward prediction errors (Caudate Nucleus), formation of memories about past events (Hippocampus), selection of action sets (SFGmed/dmPFC)[57] and processing of symbolic information related to monetary value (Angular gyrus). Since the specific aim of the current study was to shed light on how modality-specific and modality-general valuation is coordinated across the frontal and sensory areas, we only included the whole-brain activations that were located in early visual or auditory areas in our subsequent effective connectivity analyses.

## Modality-general and modality-specific stimulus value representations in the frontal cortex (multivariate results)

To evaluate the robustness of our findings and compare them with prior research[40], we conducted a multivariate pattern analysis (MVPA). This analysis is exploratory as it was not part of our initial study plan. We

performed two types of MVPA, following approaches similar to previous studies[40,58,59]. The first analysis used anatomical ROIs from the AAL atlas (lateral and posterior OFC and vmPFC; see Fig. S7) and calculated the accuracy of MVPA classifiers built from the activation patterns of all voxels within each ROI (Fig. 4A, B). The second analysis was a searchlight classification, where classification accuracies were iteratively calculated for all voxels in the frontal cortex (Fig. S3). In both cases, at the first-level analysis for each participant, we computed the accuracy of classifying trials by their respective levels of subjective value. Since the subjective value estimated from our computational model is on a continuous scale, we binned the data into four distinct levels to facilitate classification. At the group level, classification accuracies were tested against chance level (see Methods for details).

Four types of classifiers were constructed to decode the value following a previous study's approach[40]: 1) Classifiers trained and tested on auditory trials (*Aud_on_Aud*), 2) classifiers trained and tested on visual trials (*Vis_on_Vis*), 3) classifiers trained on auditory trials and tested on visual trials (*Aud_on_Vis*), and classifiers trained on visual trials and tested on auditory trials (*Vis_on_Aud*). First, we confirmed that our ROIs (vmPFC and OFC) could accurately represent the subjective value of both visual as well as auditory stimuli by examining classifiers trained and tested on the same stimulus option (e.g., *Vis_on_Vis* trained and tested on intra-modal visual red stimuli). These are denoted as *Visual* and *Auditory* classifiers in Fig. 4A, B. Next, we tested whether vmPFC and OFC could identify the value of different stimuli from the same sensory modality. For this, we inspected *Aud_on_Aud* and *Vis_on_Vis* classifiers trained and tested on different stimuli within the same modality. These classifiers were averaged across both modalities (*Aud_on_Aud* and *Vis_on_Vis*) to represent *cross-identity* classification accuracy. Finally, to discern whether our ROIs represent stimulus value independent of sensory modality, we examined the *cross-modality* classifiers, averaging the classification accuracies of *Vis_on_Aud* and *Aud_on_Vis* classifiers.

Using this approach, we found that in both vmPFC and OFC anatomical ROIs, subjective value was decoded significantly above chance levels for both visual and auditory options, confirming the role of these areas in representing subjective value. Additionally, both areas demonstrated above-chance cross-identity classification, indicating they could identify the value of different stimuli from the same modality (see cross-identity classifiers in Fig. 4A, B). Importantly, only in the vmPFC did cross-modality classifiers perform significantly above chance levels. This suggests that classifiers trained with one modality could successfully discriminate the value of stimuli from another modality in the vmPFC but not in the OFC, corroborating our univariate results.

To elucidate whether modality-general representations are exclusive to the vmPFC or they exist in other regions of the frontal ROI shown in Figs. S3 and S7, we next examined the searchlight MVPA results (Fig. 4C). Regions responsive to subjective value irrespective of sensory modality were defined as those exhibiting significant activations for the cross-modality contrast. Notably, this contrast showed activations exclusively in the vmPFC, corroborating the univariate results and previous findings[40]. The lack of cross-modality activations in the OFC despite its involvement in valuation (Fig. 4A, B) supports the idea that this region represents value in a modality-specific manner. Collectively, the univariate and multivariate results provide strong evidence for modality-specific representations in the OFC and modality-general representations in the vmPFC.

**Modality-specific effective connectivity between sensory and valuation areas**

We next examined the effective connectivity (EC) of a network consisting of sensory and valuation regions that showed significant value-related modulations in our univariate analysis. This network comprised 5 ROIs as shown in Fig. 5 (i.e., auditory and visual sensory areas: *audSen* and *visSen*; distinct modality-specific auditory and visual SVRs in the OFC: *audOFC* and *visOFC*; and overlapping modality-general SVRs in the *vmPFC*, see Methods for details). The EC analysis[60] provides an estimation of the degree

to which different connectivity patterns across this network contribute to the generation of value representations. Importantly, the EC analysis allows for an additional test of our main hypothesis regarding the existence of modality-specific representations of stimulus value across the brain, as models with and without a specific connectivity pattern for auditory and visual values were tested against each other (see the model space in Fig. 5A). We then looked for the most probable connectivity pattern in this network using a Bayesian model comparison approach[61].

In the value task, our findings revealed that the most probable model was one that included modulatory connections between the sensory cortices, modality-specific clusters in the OFC, and vmPFC (i.e., model 6, Fig. 5A, B). The winning model in the value task contained two distinct valuation sub-networks: an auditory sub-network comprising audSen, audOFC and vmPFC, and a visual sub-network with visSen, visOFC and vmPFC as nodes (Fig. 5C). Moreover, the sensory cortices did not directly communicate with vmPFC (models 2, 4, 5, 7, 8 and 10) and we did not find evidence for the cross-modality of the connectivity between the sensory cortices and the value regions in OFC (models 1, 3, 4, 5, 9 and 10, containing connections between visual cortex and auditory OFC and auditory cortex and visual OFC). Additionally, in the control task, the null model, which lacked any modulatory connections between ROIs, yielded the best fit to the data suggesting that the connectivity patterns identified by the winning model in the value task were indeed specific to the processing of the subjective value rather than the sensory properties of the stimulus options. These findings provide compelling evidence for the existence of modality-specific communication pathways which convey value-related information across the brain.

In order to understand how reward value modulates the communication of information across the brain areas, we next examined the strength of modulatory connections in the winning model of the value task. We found that during both intra-modal and inter-modal trials, all modality-specific connections were significantly modulated at a posterior probability $P > 0.99$ (Fig. 5C). Notably, we identified a clear distinction between the weights of feedforward and feedback connections. The directed feedforward connections from sensory ROIs to OFC ROIs and from OFC ROIs to vmPFC exhibited negative connection weights, signifying inhibitory modulatory connections. In contrast, the feedback connections displayed positive weights, indicating excitatory modulatory connections (for parameters of intrinsic connections and driving inputs, refer to Tables S7-S8).

Additionally, to test whether effective connectivity between any two nodes of the network was better explained by unidirectional connections, we estimated all possible unidirectional variants of the winning model (i.e., model 6 shown in Fig. 5C). As a modulatory connection between two nodes can exist in three possible ways: directed, reciprocal, bidirectional, the total number of all possible unidirectional models for the model 6 of the value task were 9, shown in Fig. 6A. The most likely model among these was the network with bidirectional connectivity between the nodes (see the model exceedance probabilities, Fig. 6B).

Collectively, the effective connectivity results showed that auditory and visual sensory cortices communicate with separate clusters in OFC, which contain modality-specific stimulus value representations (SVR) corresponding to each sensory modality. Further, the modality-specific SVRs in OFC were linked with the modality-general SVR in vmPFC to guide the final choice.

## Discussion

In order to generate specific predictive signals for adaptive goal-directed choices, the brain must encode information about the sensory modality of reward-predicting stimuli as well as the most recent value associations with the stimuli. Moreover, to be able to compare and choose between stimuli with fundamentally distinct sensory features, general value representations are equally important. Here, we used stimuli from auditory and visual sensory modalities as reward-predicting cues in a value-based decision-making context with a dynamic foraging paradigm, enabling us to identify modality-general and modality-specific value representations using

**Fig. 4 | Multivariate pattern analysis (MVPA) results. A, B** MVPA results in the anatomical ROIs, defined based on the AAL atlas. Each boxplot represents the average classification accuracies for classifying the data into four levels of subjective value across all classifiers of the same type against the chance level (i.e., the accuracy of classifiers trained with shuffled labels was subtracted from each classifier's accuracy). *Visual* and *Auditory* classifiers are a subset of *Vis-on-Vis* and *Aud-on-Aud* classifiers that were trained and tested on the same stimulus option, from either visual or auditory modality, respectively. *Cross-identity* classifiers were trained and tested on different stimuli from the same sensory modality. Cross-modality classifiers were trained on data from one sensory modality and tested on data from another sensory modality. **C** Searchlight MVPA results for the cross-modality contrast (*Vis-on-Aud + Aud-on-Vis*) shown at the peak of the depicted cluster ($P_{FWE} = 0.05$, k = 10, MNI coordinates: x = 6, y = 52, z = -6). **D** The average classification accuracy level of voxels within the cluster shown in **C** is displayed for illustration purposes. In **A, B,** and **D**, individual dots represent the data of individual participants. In **A, B,** *** indicates $P < 0.001$, ** indicates $P < 0.01$, and * indicates $P < 0.05$ for a comparison against the chance level (paired t-tests), corrected for multiple comparisons (Bonferroni correction). Chance level in all panels is calculated based on classifiers trained with shuffled class labels.

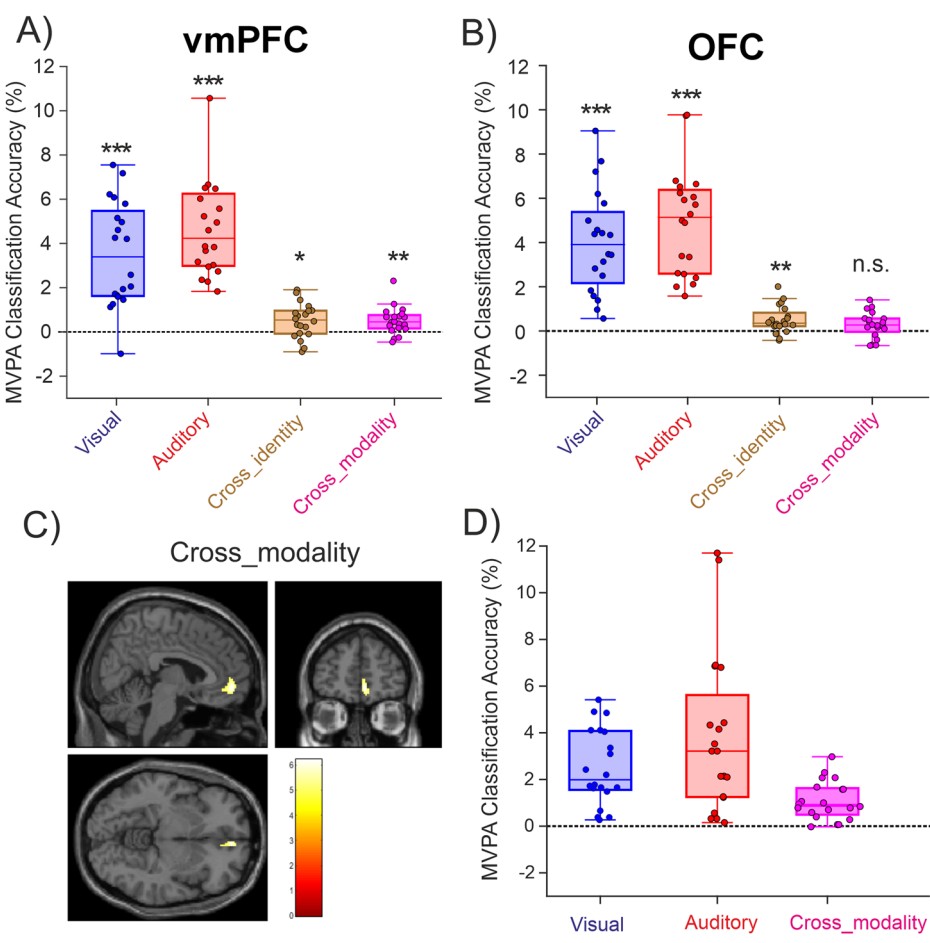

univariate fMRI analysis and reveal the underlying neural mechanisms that generate the two types of representations using effective connectivity analyses.

We found trial-by-trial value representations of auditory and visual sensory modalities to be present in segregated lateral and posterior regions of OFC, respectively. This effect cannot be due to differences in reward sensitivity or difficulty of choices since neither the choice ratios relative to rewards nor the reaction times indicated a difference between auditory and visual conditions. We also did not observe any difference in fit parameters of our computational model across sensory modalities. Furthermore, we verified using the control task that the segregation in modality-specific representations is not due to the differences in sensory processing mechanisms underlying the auditory and visual sensory modalities. These results were corroborated by an exploratory multivariate pattern classification (MVPA) analysis showing that only in the vmPFC but not in the OFC stimulus value could be decoded independently of the sensory modality. Thus, our convergent findings from multiple types of analysis show for the first time the presence of dedicated neuronal populations in OFC that reflect updates in value associations of a particular sensory modality and generate specific predictive signals. As such, the present findings are in line with the known role of the OFC in representing a "cognitive-map" of the task space[23,24], especially when a task involves reversal learning [24,62] or devaluation of previously valuable options[63]. More specifically, these findings support recent proposals that the representation of stimulus value is an active hierarchical process in which the OFC plays a key role in representing the value of individual features of a stimuli[39,64].

In contrast to the modality-specific value representations found in OFC, we found modality-general value representations in vmPFC. Specifically, auditory value representations were found to overlap with visual

value representations, aligning with the concept of the vmPFC as a common currency coding hub for distinct categories of rewarding stimuli[7-9,11,32-37], a finding that was also confirmed in our MVPA analysis. However, subdivisions from general to specific valuation have been also shown in vmPFC in the anterior-to-posterior direction[32,35,65-67], where anterior vmPFC represents values of distinct reward categories in a general manner and posterior vmPFC in a specific manner. The loci of overlapping activations that we found in this study were in anterior vmPFC and thus support a role of the anterior vmPFC in common currency coding of value. However, we did not find any modality-specific value representations in vmPFC (even in the posterior part), which may either be due to OFC being exclusively responsible for implementing modality-specificity in a task such as ours, or be related to the specific type of reward category, i.e., monetary rewards, that we employed in our task[32]. Interestingly, we also found that vmPFC activations were even present in the control task where no continuous and gradual value-related information processing was needed. This observation points to a general role of vmPFC in computation of choice in addition to valuation, as suggested by recent theoretical frameworks[68].

Whereas the majority of previous studies have underscored a common currency coding of subjective value, few recent studies have provided evidence for the stimulus-specific (or identity-specific) representations of reward-predicting visual stimuli or specific reward outcomes[19,30-32,38], as well as modality-specific valence[69]. We are only aware of one neuroimaging study by Shuster and Levy[40] where modality-specificity of reward-predicting valuation across auditory and visual domains was examined. Using a risk-based choice task where monetary values of lotteries were either presented visually or announced aurally, this study[40] found that the anterior portion of vmPFC represents value irrespective of the sensory modality, whereas no evidence for modality-specificity was found beyond sensory areas. Several

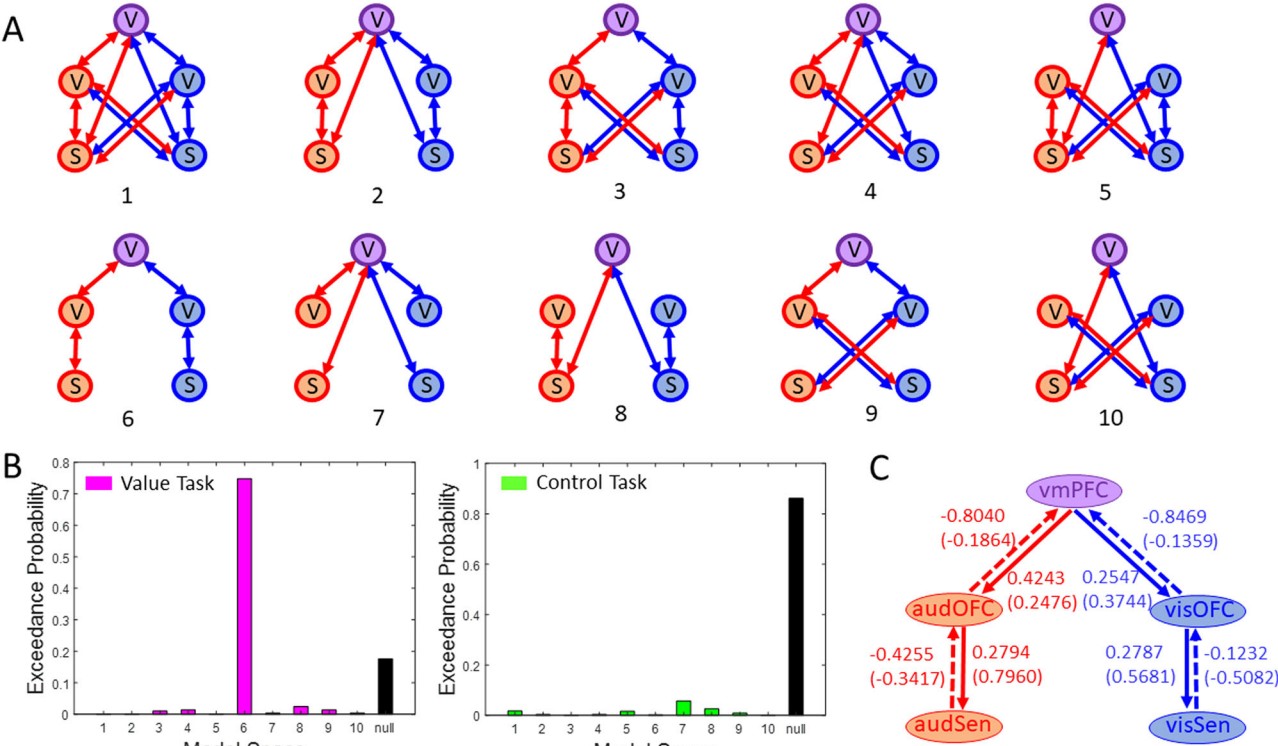

**Fig. 5 | Modality-specific effective connectivity. A** Model space consisted of 10 biologically plausible models per task (value or control). Models differed in their modulatory connections between nodes of a network comprising 5 ROIs (ROIs: V – Valuation region, S – Sensory region, purple V – vmPFC, red V – Auditory OFC, blue V – Visual OFC, red S – Auditory Sensory Cortex, blue S – Visual Sensory Cortex, see also **C**). Modulatory connections in red exist during conditions in which an auditory stimulus was selected and those in blue exist during conditions in which a visual stimulus was selected in both intra- and inter-modal conditions. **B** Exceedance probabilities for all connectivity models in the value and control task.

In the value task, Model 6 is the most likely model, with exceedance probability of 0.75. In the control task, the null model, which did not have modulatory connectivity between ROI pairs for any condition was the winning model. **C** The weights of feedforward (dashed) and feedback (solid) modulatory connections in the winning model of the value task. Connection weights are shown for conditions in which an auditory (red) or a visual (blue) stimulus either in intra-modal or inter-modal conditions (in brackets) was selected. All parameters were significant at posterior probability of $P > 0.99$.

factors could explain the absence of modality-specificity in the OFC in Shuster and Levy's study[40]. Firstly, their design involved visual and auditory stimuli (text representation and audio recording of lottery numbers) that could be directly and unequivocally translated into explicit numeric values. This contrasts with our study, where specific sensory features of choice options (red/green checkerboard circles and low/high pitch tones) had to be disambiguated in terms of their link to rewards. Secondly, in our design, distinct features of choice options had to be continuously tracked across trials, and their subjective value had to be updated based on choice and reward history, a key difference from previous studies[40]. Given that one of the major roles of the OFC is updating the valuation of options based on past reward outcomes[24], this could explain the discrepancy between the two studies. Due to the importance of tracking sensory features of stimuli in our design, we needed to account for the covariation of sensory and reward processing[18]. For this, we used a control task which was similar in sensory requirements and final choice to the value task but did not require updating computed values for each sensory modality. These features of our design allowed us to uncover both modality-specific and modality-general representations of value in the frontal cortex and extend previous findings[40].

Apart from the frontal cortex, we found value modulations in sensory cortices, which provide further evidence that representations of value are not restricted only to higher cognitive areas, as has been shown before[50]. The value representations in sensory cortices were largely modality-specific, which means that individual sensory cortices represented the value of stimuli presented in their own sensory domain, as has been previously shown for the representation of value[40] and valence[69]. These findings raised interesting questions regarding whether and how a communication of value-

related information exists between sensory cortices and valuation regions. Interestingly, we found that the auditory and visual sensory cortices were bi-directionally connected to the lateral and posterior OFC (corresponding to auditory and visual value representations), respectively, in a modality-specific manner. Specifically, the modality-specific effective connectivity results revealed a high degree of selectivity: in trials when planning to choose auditory reward stimulus, there was a significant connectivity from the auditory sensory cortex to lateral OFC for these trials and not otherwise. A similar modality-specific significant connectivity existed from visual sensory cortex to posterior OFC when choosing visual reward-predicting stimulus. This finding is in line with a previous work showing connectivity between OFC and piriform cortex (relevant in case of odour stimuli) for the formation of stimulus-specific value representations in OFC[31]. Moreover, past studies have shown that lateral and posterior regions of OFC receive direct afferent inputs from auditory and visual sensory cortices[41,70], providing neuroanatomical support for our findings.

The connectivity between sensory cortices and modality-specific representations in the OFC allows the frontal valuation areas to have access to sensory representations of rewarding stimuli in each sensory area. The sign of this connectivity provides additional insights into how modality-specific valuation is implemented. We found that feedforward connectivity in modality-specific networks was inhibitory. Feedforward inhibition has been suggested as a key mechanism in imposing temporal structure to neuronal responses[71] and expanding their dynamic range of activity[72]. These mechanisms allow the OFC to form an integrated representation of the sensory and value information across time, rather than encoding the precise sensory features of stimuli at every moment (akin to sensory

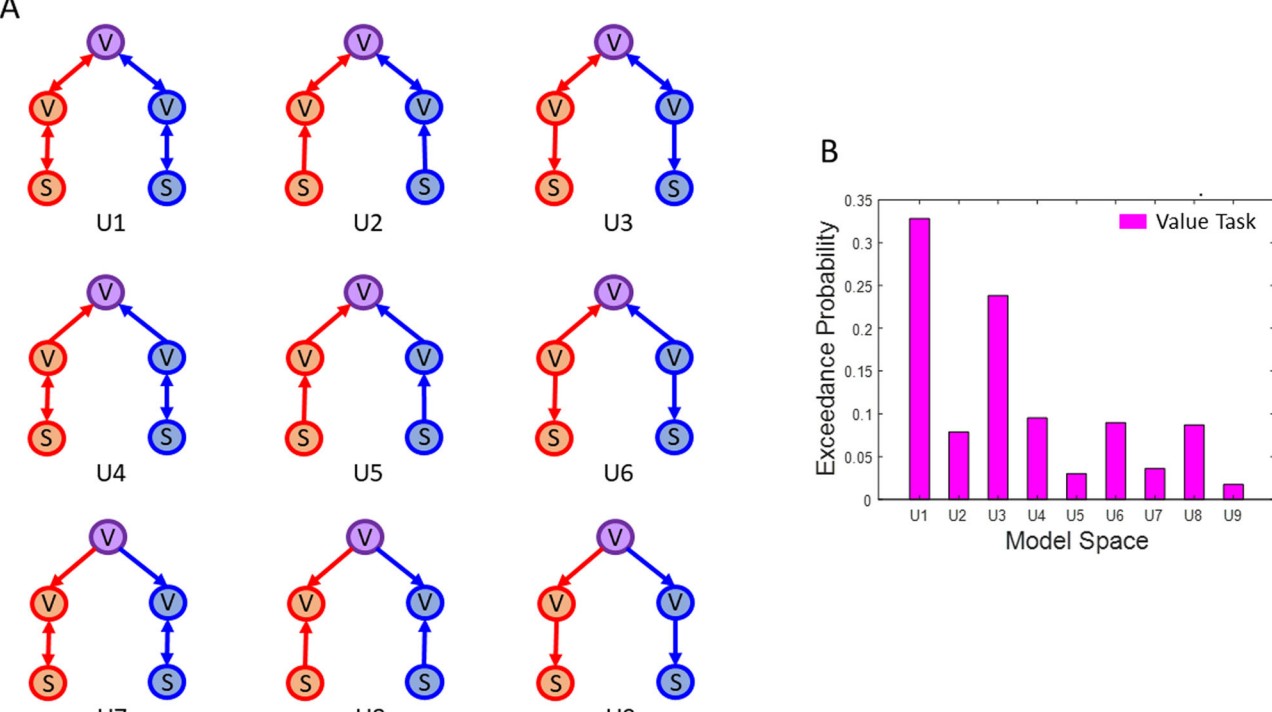

**Fig. 6 | Uni- and bi-directional variants of the winning model in the value task.** **A** Model space consisting of all possible unidirectional variants of the winning model in the value task (i.e., U1, also refer to the legend of Fig. 5A and the schema in Fig. 5C for information on nodes and modulatory connections). **B** Exceedance probabilities for the model space in **A**. Model U1, i.e., a model containing bidirectional connections between all nodes, is the most likely model (ROIs: V – Valuation region, S – Sensory region, purple V – vmPFC, red V – Auditory OFC, blue V – Visual OFC, red S – Auditory Sensory Cortex, blue S – Visual Sensory Cortex). The second best model is U3, which only contained feedback connections between the OFC nodes (red and blue V) and sensory cortices (red and blue S).

cortices), thereby serving as a cognitive temporal map of the task[23,24]. Additionally, connectivity results showed that the value modulations in sensory cortices were driven by top-down feedback signals generated in respective valuation regions in OFC. This is consistent with previous work showing that biasing signals generated from frontal and parietal areas modulate spatially selective visual areas[50]. In fact, recent studies have provided robust causal evidence for the role of lateral OFC in value-driven guidance of information processing in sensory cortices[73]. Our finding of the presence of excitatory feedback connectivity between the modality-specific representations in lateral and posterior OFC and auditory and visual cortices, provides strong support for the causal role of top-down valuation signals in shaping sensory perception during decision-making.

Further, we found that specific value representation in OFC were linked to general value representations in vmPFC. Specifically, we showed that when planning to select an auditory reward-predicting option, there was a change in the connectivity between the auditory value representations in OFC and modality-general representations in vmPFC, with a similar pattern found for the selection of the visual options. This result highlights the underlying mechanism whereby value representations in OFC provide input to the vmPFC to support the formation of general value representations needed for the comparison of options from distinct modalities and deriving the final choice. Notably, the modality-specific connectivity between OFC and vmPFC corroborates previous findings showing that sensory-specific satiety-related changes in connectivity between OFC and vmPFC predicted choices in a devaluation task[30]. Together, these results show how common currency coding of value integrates modality-specific information about reward-predicting stimuli in a dynamic environment to guide choices. However, it is crucial to interpret the effective connectivity results with caution. For example, as illustrated in Fig. 6, various versions of the winning model, differing in their mono- versus bi-directional connectivity, produced similarly good fits to the data. Therefore, drawing definitive conclusions about the specific modes and directions of

connectivity across the networks shown in Figs. 5 and 6 remains challenging at this stage. Nevertheless, our study provides a valuable framework for exploring how the representation of value and sensory features across the brain underpins valuation.

Understanding whether valuation signals in frontal cortex contain information about the sensory modality of reward-predicting stimuli has a number of important theoretical and clinical implications that go beyond the specialized field of neuroeconomics and value-based decision making[1,13,74]. We show that value-based choices involving reward-predicting stimuli from different sensory modalities are supported by connectivity between the sensory areas and the modality-specific representations in OFC. Although top-down modulation of perception through interactions between frontal and sensory areas has been the basic tenet of a number of influential theoretical frameworks[75–78], the importance of modality-specific representations of reward value in frontal areas that could provide a biologically plausible implementation of these putative interactions has been largely ignored. Therefore, our study provides novel insight for future computational work on how top-down signals can be selectively routed to impact on sensory processing. In doing so, it is important to note that the modality-specific representations that we found may adapt and reorganize under different contexts rather than being hardwired and fixed in the brain see also ref. 59. In fact, outcome-related adaptation in the representation of value can occur during the same task[79], which provides a flexible mechanism for reorganizing neuronal codes of value based on the context. Future studies will be needed to examine whether and to what extent the modality-specific coding of value can adapt to the specific features of a task. From a clinical perspective, our results suggest that localized lesions to OFC may be associated with specialized impairments of value-based decisions in visual or auditory domains, an interesting possibility that can be further investigated by future studies. Additionally, our findings may allow a better understanding of pathological states such hallucinations[80,81] where illusory percepts arise in the absence of external stimuli[82], likely due to the aberrations in

https://doi.org/10.1038/s42003-024-07253-8    **Article**

communication pathways between the frontal and sensory areas[83]. More generally, the present study, together with previous efforts in understanding how value-related information is communicated between the frontal and sensory areas[31,73,84], provide instrumental insights regarding how perceptual and cognitive processes are coordinated in the brain.

In summary, our results provide novel evidence for the co-existence of modality-specific and modality-general encoding of subjective value in OFC and vmPFC, respectively, pointing to the specialized functions of these two valuation areas. A general value signal would facilitate the comparison between distinct rewarding stimuli[7,13] and the transformation of stimulus values into motor commands[37]. On the contrary, modality-specific value encoding associated to respective sensory cortical representations would support goal-directed adaptive behaviour by generating specific predictive signals about impending goals[20,22,23], such as when planning to choose auditory or visual reward-predicting stimuli. We further show how the communication between sensory areas and modality-specific representations of subjective value in OFC plays a central role in supporting value-based decisions in a multimodal dynamic environment.

## Methods

### Participants

Twenty-four healthy subjects (13 male and 11 female, age 19 to 45 years; mean ± SD age = 27.92 ± 6.04 years, including one of the co-authors) participated in the experiment for financial compensation of 8€/hour. The sample size was based on a previous study that used a similar paradigm[50]. Furthermore, to ensure adequate statistical power for detecting valuation across different sensory modality configurations, each participant underwent scanning during two sessions, thereby providing a sufficient number of repetitions for each configuration (see Experimental Design). Each session lasted about 150 minutes (comprising 45 min preparation time and 105 min scanning time: 90 min functional and 15 min structural scans). Before the first session an online training (30 min) was scheduled to familiarise participants with the task. Participants also had the opportunity to earn a monetary bonus of maximum 22€ based on their behavioural performance in the value-based decision-making task (value task) during the scanning session. All participants were right-handed and had normal or corrected-to-normal vision. Before the experiment started and after all procedures were explained, participants gave an informed written consent and participated in a practice session. The study was approved by the local ethics committee of the "Universitätsmedizin Göttingen" (UMG), under the proposal number 15/7/15. All ethical regulations relevant to human research participants were followed.

Four participants were excluded from the final analysis resulting in the data from 20 subjects presented here (10 male and 10 female, age 21 to 42 years; mean ± SD age = 29.00 ± 6.34 years): two participants had difficulty in differentiating the strategies of the value and the control task (specifically with the instructions associated with the feedback colours in the two tasks, see Experimental Design); one participant was excluded due to excessive head motion while scanning ( > 4 mm); and one participant due to the unusually large size of the ventricles in the structural MRI scan.

### Experimental design

The experiment consisted of a value-based (value task) and an instruction-based (control task) decision-making task, completed in two sessions (Fig. 1). Each session consisted of 12 blocks (of 72 trials each): 9 blocks of the value task (i.e., 3 blocks for each of the 3 reward ratios; 1:3, 1:1, 3:1; for details see the *Dynamic Reward Structure*) followed by 3 blocks of the control task. Each task involved a binary choice between stimuli presented in one or two sensory domains: both auditory (*AudAud*), both visual (*VisVis*), and audio-visual (*AudVis*), which were presented in separate blocks. All three types of sensory conditions appeared an equal number of times across each task in a pseudo-random order.

**Stimuli**. Two pure auditory tones (low pitch (LP) tone sawtooth, 294 Hz; high pitch (HP) tone sinusoidal, 1000 Hz, played through MR-compatible earphones - Sensimetric S15, Sensimetrics Corporation, Gloucester, MA - with an eartip - Comply™ Foam Canal Tips) and two contrast reversing visual checkerboards (green and black or red and black with the contrast reversing at 8 Hz, as in ref. 50) within circular apertures (4 ° radius) were used as the choice options. In an auditory (*AudAud*) or a visual (*VisVis*) trial, either two tones or two checkerboards were presented as options, respectively. In an audio-visual (*AudVis*) trial, one tone and one checkerboard were presented as options. Choice options were presented simultaneously on the left or right side of the centre (auditory stimuli were played one on each side of the earphones; visual stimuli were centred at the 10 ° horizontal distance and 5 ° above the centre of the screen). Different tones and coloured checkerboards and their combinations (in *AudVis* blocks) were presented an equal number of times across the 72 trials of a block in a pseudorandom order.

**Trial structure**. Both the value and control tasks featured identical presentations of stimulus options, response requirements, and feedback on the decisions. The only distinction between them lay in the cues associated with the feedback colours (Fig. 1A, B). Participants were asked to fixate continuously throughout each run (here, a run = 3 blocks) on a small square (0.4 ° visual degree) at the centre of the screen. A trial began with a mean fixation period of 1.8 s (±0.45 s), yielding a mean trial duration of 4.3 s. Following the fixation period, the two stimuli options were presented simultaneously for 1 s, one on each side of fixation. The spatial position of each option was also pseudo-randomised across the trials of a block in such a way that each option appeared an equal number of times on both sides of the fixation point. Following the onset of the stimulus options, participants pressed either the left or the right button on a MR-compatible two-buttoned response box (Current Designs Inc., Philadelphia, PA), using the index or the middle finger of their right hand, to indicate their choice. The participants were required to respond within 2.25 s following the onset of the options. Following the response window, a feedback window of 0.25 s appeared in which the central fixation point turned either yellow or blue in colour. In the value task, the yellow fixation indicated that the choice was rewarded, and the blue fixation indicated that the choice was not rewarded. Since the control task was designed to be similar to the value task in terms of sensory processing requirements without a need to track and update their estimation of options' value, the feedback instructed the participant to make a prescribed choice. Thus, in the control task, the yellow fixation indicated to switch from the past trial choice and the blue fixation indicated to keep the past trial choice. The choice on the first trial of the control task in each block was a random choice.

Note that in all intervals during a trial in both tasks, two placeholders (circular apertures: 4 ° radius) containing white noise were presented on the screen. This was specifically done to reduce the visual after-effects which could ensue from the presentation of the coloured checkerboards, and additionally minimized the low-level sensory differences between the different stimulus configurations. When choice options were from the visual modality, the placeholders were replaced with the coloured checkerboards and otherwise stayed in view on the screen (Fig. 1A).

**Dynamic reward structure**. To create a dynamic multimodal environment for participants, rewards were assigned to the options independently and stochastically at random intervals using a Poisson process[51]. On average, a reward was available for delivery on 33% of the trials (of a block of value task). These 24 rewards in a block (33% of 72 trials) were distributed between the two stimuli options in different reward ratios of {1:3, 1:1, 3:1}, such that the rewards assigned to options were {8.5%:24.5%, 16.5%:16.5%, 24.5%:8.5%} in percentage of trials. For the value task in a single session (9 blocks), these three reward ratios were repeated and randomised such that each reward ratio was used exactly once with every sensory domain block (i.e., *AudAud*, *VisVis*, and *AudVis*). The randomisation of various factors such as sensory modality, spatial position of options, and reward ratios was done to provide a

dynamic environment, in which the participant would be required not only to update their stimuli-value associations with changing reward ratios but also to keep track of the reward-predicting stimuli very carefully on a trial-to-trial basis as spatial positions changed. Two important schemes of "baiting" and "change over delay" (COD) were adopted as in previous studies[50,51]. Firstly, the assigned reward to an option was baited only until that option was chosen. Baiting was introduced to discourage an "extreme exploitation" strategy in which a participant would always stick to the option with a higher reward rate (e.g., 24.5% > 8.5%) association in a block and to motivate exploration of both options. Secondly, an earned reward feedback was delayed for one trial when the participant changed their choice from one option to the other and delivered only if the participant chose the same option again. This cost, i.e., COD, was employed to discourage "extreme exploration" strategy, where the participant would be able to consume all rewards without any learning by alternating choices rapidly between options. Trials following a change of choice (switch) between options were not included in the analysis because subjects were informed that they will not get a reward on such trials and hence choices were not completely free. At the end of each block, participants were shown the reward earned in that block at the rate of 5 cents per yellow square shown as the reward feedback. At the end of the second session, participants received the total reward earned, which was up to a maximum of 22€ (11€ per session) based on their performance along with a participation fee of 8€ per hour.

**Control task structure.** Similar to the reward structure in the value task, switches were assigned independently and stochastically to the options in an equiprobable manner with an average switch rate of 33%. Thus, on any trial when a participant earned a switch from a chosen option, yellow feedback was displayed indicating that they should switch their choice to the other option on the next trial. On other trials, when a switch was not assigned, blue feedback was shown to indicate that the same option should be chosen on the next trial. This type of switch assignment structure was developed to encourage a similar temporal choice pattern as in the value task. To confirm that this design met its intended purpose, we examined participants' choices in the value task and found an average switching rate of 27.12% ($\pm$ 0.02% s.e.m.), closely aligning with the switch rate used in the control task. On a single day, the control task was conducted in each of the three sensory domains. There were no baiting and COD schemes employed in the control task. At the end of each block, participants were shown their performance that indicated how accurately they followed the instructions in that block.

**Computational framework of choice behaviour**
To examine whether participants' choices in the value task were influenced by the dynamic reward structure employed in our design, we used a computational framework that has been used in the past to model choice behaviour abiding by the matching law[50–52]. In our task, there were no prior reward associations with the options, and hence on any trial $t$ a participant made a choice $c(t)$ based on the previous rewards received $r(t^-)$ during the experiment, see Fig. 1C. Intuitively, an option that delivers more rewards per unit of time should have relatively higher value associations and should be chosen more often. Thus, to estimate participants' subjective value beliefs for each reward option on a trial-by-trial basis, we fitted the reward history and choice data of each participant to a linear-nonlinear-probabilistic (LNP) model, shown in Fig. 1C (also called as linear regression-based model of reinforcement learning[85]). We chose an LNP model over other reinforcement learning models (RL models such as Q-learning) since the former has been shown to better capture the statistics of the matching behaviour that is observed in our paradigm[51]. Two broad phases of the LNP model are the *learning* and the *decision-making* phase[50,51].

In the *learning phase* (see Fig. 1C), two identical linear filters ($n$ learning weights $\alpha_\tau$, $\tau = 1$ *to* $n$ trials in the past) weigh the reward history (till $n$ past trials) of each option ($r_i(t^-)$, $i = 1, 2$ corresponding to stimulus sets $S_1, S_2$),

where $n$ is equal to half of the trials over which the reward ratio was unchanged (here, $n = 36$)[50]. The purpose of the filter is to look back on the reward history of past $n$ trials and distil the effect of those trials into a scalar value. This scalar value should be representative of the participant's expectations for associated value of stimulus options. In other words, if the choice of a particular option was rewarded (or not rewarded) on the past trials, then the value belief for that option should be relatively higher (or lower) on the current trial, respectively.

First, we explain how filter weights were derived from the reward history and choice data of a participant and then how these weights were used to compute the subjective value beliefs for the stimuli options on each trial. The reward history $r_i(t^-)$ of a stimulus option $S_i$ is a binary vector containing 1 when that option was chosen and rewarded on trial $t$ and 0 when chosen and not rewarded on trial $t$ or simply when not chosen on that trial. As explained in previous section on experimental design, trials following a change of choice (switch) between options were not included in the behavioural analysis because subjects were informed that they will not get a reward on such trials and hence choices were not completely free. Thus, the free choice $c_i(t)$ of a stimulus option $S_i$ is denoted by 1 when that option was chosen on trial $t$ and 0 when not chosen on that trial. As the overall reward assignment over the two options was symmetric, their impact on choice was equal and opposite, hence we used the composite reward history $r$ (as shown in (1)) and composite choices $c$ (shown in (2)) for further analysis.

$$r = r_1 - r_2 \tag{1}$$

$$c = c_1 - c_2 \tag{2}$$

A participant's choices are strongly influenced by the reward history (the learning mechanism), thus, intuitively the filter weights should most closely match the input reward history to the desired output choices. This is an optimization problem[50,51] and can be solved by employing Wiener-Hopf equations:

$$\alpha_\tau = C_{rr}^{-1} C_{rc} \tag{3}$$

where, $C_{rr}$ is auto-covariance matrix of input time series $r(t^-)$ and $C_{rc}$ is cross-covariance vector of output time series $c(t)$. Further, what we used in the analysis were relative filter weights $\hat{\alpha}_\tau$, which were obtained by normalising weights obtained in (3) such that the sum of all $n$ weights equals 1 (Fig. 2C shows relative filter weights of a single participant). For more implementation details refer to[50,51].

Next, the relative filter weights were used to compute subjective value beliefs for the stimuli options $v_i(t)$, $i = 1, 2$ on trial $t$. This was done by multiplying the reward history of option $i$ on past $n$ trials by the corresponding weighting coefficients $\hat{\alpha}_\tau$ and summing the product results over the past $n$ trials (see Fig. 1C):

$$v_i(t) = \sum_{\tau=1}^{n} \hat{\alpha}_\tau r_i(t^-) = \sum_{\tau=1}^{n} \hat{\alpha}_\tau r_i(t - \tau) \tag{4}$$

The *decision-making phase* (see Fig. 1C), draws the ultimate binary choice ($S_1$ or $S_2$) on trial $t$ based on a relation that maps the differential value $dv(t)$ (as shown in (5)) computed on trial $t$ to the participant's probability of choosing option $S_1$ on that trial. Intuitively, this relation should strongly predict a participant's choice behaviour, where the participant should make a choice $c(t)$ based on the comparison process shown in (6).

$$dv(t) = v_1(t) - v_2(t) \tag{5}$$

$$c(t) = \begin{cases} S_1, & if\ v_1 > v_2 \\ S_2, & if\ v_1 < v_2 \end{cases} \tag{6}$$

To assess the fit of the LNP model during the *learning phase*, the relative filter weights for the data of each participant were approximated by

fitting an exponentially decaying function, indicating that choices were most impacted by recent rewards rather than distant rewards in past (quantified by time scale parameter trials of the fit; see Fig. 1C for the illustration of the filter and Fig. 2C for the fit to the data of a single participant).

To assess the *decision-making phase*, i.e. the effect of the differential value on choices, we separated trials of the value task into 20 bins of equal size, based on their differential value $dv(t)$ (see Eq. (5)). The probability of choosing option $S_1$ in a bin was obtained by computing the proportion of $S_1$ choices made in the subset of binned trials. Intuitively, as the differential value between the stimuli options increases, the probability of choosing $S_1$ should also increase (see scatter plot in Fig. 2D). Thereafter, the probability of choosing option $S_1$ was approximated by fitting a normal cumulative distribution function (Eq. (7)).

$$\varphi(x, \mu, \sigma) = \frac{1}{2\pi\sigma^2} \int_{-\infty}^{x} e^{-\frac{(x-\mu)^2}{2\sigma^2}} dx \qquad (7)$$

where x is the differential value ($dv$). This function contains two important decision-making parameters: $\mu$ corresponding to a participant's biasness towards a particular option and $\sigma$ that measures the sensitivity to value differences or in other words the explore-exploit tendency. Accordingly, $\sigma = 0$ corresponds to an extreme exploitative tendency, and $\sigma \to \infty$ to extreme exploration. The disadvantage of being extremely exploitative; i.e., sticking to an option that has higher reward rate associated with it, is that it would yield lesser number of rewards to the participant because there exist unvisited options, which remain baited until chosen. Moreover, extreme exploration would also be disadvantageous, as it would lead to no learning and the absence of any strategy. Thus, the optimal strategy in this task would be to choose more often the option with higher reward rate and to occasionally visit the less rewarding option to consume rewards on it. An optimal strategy is advantageous in a dynamic reward structure task where the aim is to maximize rewards, and to examine whether this is the case in our task, we inspected the abovementioned parameters ($\tau, \mu, \sigma$) for their fit to participants' behavioural data (Fig. 2C-E).

In the value task, the positive and negative feedbacks have distinct effects on participants' beliefs. Therefore, if the choice of a particular option was rewarded (yellow feedback) or not rewarded (blue feedback) on the previous trial, then the value beliefs for that option should be relatively higher or lower, respectively, on the current trial in comparison to the value beliefs in the past trial. As only one of the two options could be chosen and rewarded in any trial, the differential value of two options would also be relatively high in magnitude when reward was received on the past trial, and otherwise low. On the contrary, the control task was designed in a way to be like the value task in terms of sensory processing requirements but not involve the participant in any learning or updating of the stimuli value. Intuitively, when no learning via feedbacks occurs, the two types of feedbacks (keep/switch) should have a similar effect on the subjective preference over the options. To confirm this, we tested the fit of the same LNP model to the choices in the control task and compared the absolute differential values of each trial obtained from models' fits to both tasks (value and control tasks) against the type of feedback received (blue or yellow) in the previous trial (Fig. 2F). We used the absolute differential values ($absDVs$) as a measure of subjective preferences because the choice behaviour is symmetric with respect to the individual options. Additionally, we also compared the fit of the model estimated from the value task to the test data from the value task (leave-one-out procedure as done in a previous study[50]) and control task (fit parameters derived from all data in the value task and tested on the control task).

## fMRI data acquisition and pre-processing
MRI scanning was carried out on a 3-Tesla Siemens MAGNETOM Prisma scanner equipped with a 64-channel head-neck coil at the Universitätsmedizin Göttingen. Anatomical images were acquired using an MPRAGE T1-weighted sequence that yielded images with a $1 \times 1 \times 1$ mm resolution. For fMRI, whole-brain gradient-echo echoplanar imaging (EPI)

volumes were acquired in transverse orientation (TR = 1500 ms, TE = 30 ms, voxel size = $2 \times 2 \times 2$ mm, flip angle = 70°, image matrix = $104 \times 104$, field of view = 210 mm, slice thickness = 2 mm, 0.2 mm gap, PE acceleration factor = 3, 69 slices, multi-band acceleration factor = 3, GRAPPA factor = 2). Data from each participant was collected in two identical sessions on two separate days. An experimental session consisted of multiple runs of fMRI data acquisition, where a run comprised starting the scan and acquiring data for three blocks of the tasks ( ~ 16.3 minutes) after which the scan was stopped and resumed again after a break ( ~ 5 minutes). To account for the signal dropout in the frontal regions, we used a data acquisition protocol that minimizes signal dropout in this region[86]. This was achieved through using a PA phase encoding direction in combination with a rather high in-plane spatial resolution and rather thin slices (2 x 2 x 2 mm) for our EPIs. For every fMRI run, we also acquired a field map, which allowed us to have a close match between fMRI and field maps used to correct image distortions during the preprocessing. Using these measures, we successfully mitigated the signal drop-out in the frontal areas (see Fig. S8). On each day, four fMRI runs (first three runs: 9 blocks of the value task, last run: 3 blocks of the control task) were conducted.

Data pre-processing and further statistical analyses were performed using Statistical Parametric Mapping software (version SPM12: v7487; https://www.fil.ion.ucl.ac.uk/spm/) and custom time-series analysis routines written in MATLAB. EPI images of each session were slice time corrected, motion corrected, and distortion corrected by using the measured field maps. The T1 anatomical image was co-registered to the mean EPI from realign-&-unwarp step, and then segmented. The estimated deformation fields from the segmentation were used for spatial normalisation of the corrected functional and anatomical images from the native to the MNI (Montreal Neurological Institute) space. Finally, the normalised EPI images were spatially smoothed using a $6 \times 6 \times 6$ mm FWHM Gaussian kernel.

## fMRI univariate analysis: general linear modelling (univarGLM)
For each participant, we first specified a general linear model (GLM) using the pre-processed functional images of two sessions that were concatenated. The GLM modelled both the value and the control task using 35 event-related regressors (see also Table S3) convolved with the canonical hemodynamic response function (HRF). For the value task, we defined individually for each of the three modality conditions (auditory, visual, audio-visual) one unmodulated stick regressor representing the modality-wise trial identity and two parametrically modulated stick regressors containing the trial-by-trial updated subjective value (SV) beliefs regarding each of the options presented, referred to as the value-modulated regressors. Trial identity was entered as 1 at the onset of the stimuli for trials of a particular modality condition and 0 otherwise. The value-modulated parametric regressors (SVs) were estimated based on the LNP model (see Eq. 4) and represented the trial-by-trial learning and updating of subjective value of each option separately (Fig. 1C and Supplementary Fig. S4). The trial-by-trial SVs were entered at the onset of the stimuli options and are denoted by $lpSV$ and $hpSV$ in auditory domain corresponding to low pitch and high pitch tones, $rSV$ and $gSV$ in visual domain corresponding to red and green checkerboard stimuli, $aSV$ and $vSV$ in audio-visual domain corresponding to auditory and visual stimuli in any combination (see also Tables 1, 2 and S5).

Similarly, for the control task we defined individually for each of the three modality-domains one unmodulated regressor representing the modality-wise trial identity and two parametrically modulated regressors corresponding to each of the options presented. In the control task, the aim was to passively follow instructions. Thus, to create a parametrically modulated regressor corresponding to one stimulus option, a weight of either 1 or 0 was assigned at the onset of stimuli options in each trial depending on whether the instruction (keep/switch your choice) from the last trial was correctly followed or not, respectively (see also Supplementary Table S4 and Fig. S4).

Additionally, we also modelled the response to the feedback displays in each trial. Please note that in the value task, the feedback display in a trial

indicated whether the chosen option in the current trial was rewarded or not; while in the control task, the feedback indicated whether to switch or keep the current trial's choice in the next trial (for more details, refer to the Experimental Design section). In order to separate signals related to the expected value of the stimuli from the signals related to the choice (response) and receipt of an outcome (feedback), we included two unmodulated event-related regressors (collapsed across the value and the control task) locked to the time of response and the onset of feedback in our GLM. Note that in our paradigm, in any current trial '*t*', the stimuli-value associations are updated based on the feedback event in the previous trial '*t*-1'. Therefore, in our task design we had introduced a jitter between the feedback in the previous trial and the stimuli presentation in the current trial. This jitter together with modelling the feedback and response with separate repressors allowed us to separate the signals related to the expectation of stimuli value from the signals related to the receipt of an outcome, which would have been otherwise impossible due to the sluggish nature of the hemodynamic responses. Furthermore, 15 nuisance regressors were included in the GLM corresponding to the following: instruction presentation at the start of each block, six motion parameters, run regressors (modelled by assigning a weight of 1 for each volume of that run and else 0: a run corresponds to each period of MRI data acquisition between the start and the end of the scan) to account for the difference in the mean signal activity between each time the scan started (one less in number than the total number of fMRI runs, here 7) and a constant (for a complete list of regressors see Table S3).

Based on the GLM described above, the brain regions which represented subjective value in each domain were identified by inspecting several univariate contrasts of the value-modulated parametric regressors (Fig. 3 and Tables 1–3 and Figs. S5-S6 and Tables S5-S6). Note that our primary interest in this study was to identify the neural correlates of valuation for different sensory configurations. Since in a binary choice situation valuation occurs for each of the two options separately, we used the trial-by-trial value-modulations of each option as an independent variable (one parametric regressor for each option in our GLM) to identify areas in the brain which encode the value of each option. This is a different approach than using the differential value of the options as the independent variable which seeks to identify the neural correlates of comparison and choice between options[50] rather than the valuation of each individual option which occurs before the choice process. Accordingly, in intra-modal conditions (*AudAud* and *Vis-Vis*), we estimated an overall effect of value-modulated regressors separately in auditory and visual sensory domains by defining contrasts: *intraAudSV > 0* and *intraVisSV > 0*. The auditory contrast *intraAudSV > 0* is the combined effect of parametric regressors of interest *lpSV and hpSV* (according to SPM convention, the contrast vector corresponding to 35 regressors contains 1 for the two parametric regressors of interest, here *lpSV* and *hpSV*, and 0 otherwise). Similarly, for the visual contrast *intraVisSV > 0* is the combined effect of parametric regressors *rSV and gSV*. In the inter-modal condition (*AudVis*), we estimated the overall effect of value modulations in both sensory modalities using the contrast: *interAudVisSV > 0*, where *interAudVisSV > 0* is the combined effect of parametric regressors of interest *aSV and vSV*. To compare the results based on the above contrasts with those from an analysis focusing on the representation of the differential value of options, we have included the latter in the Supplementary Information (see Fig. S9, Supplementary Text, and Table S9).

Note that our GLMs included two parametric regressors each modeling one of the two choice options in a trial of value and control tasks. This inherently introduces some degree of (anti)correlation between the two parametric regressors, as in every binary choice situation the subjective value of one option is influenced to some extent by the subjective value of the other option. To mitigate this correlation, we employed two strategies. Firstly, we modeled these regressors across the entire experiment, including both value and control tasks within the same GLM. This reduces collinearity, as during intervals when participants perform a different task, the two parametric regressors are not correlated. Secondly, we utilized orthogonalization of regressors based on the standard method implemented in SPM[87]. This technique removes shared variance between the regressors due to

collinearity and assigns it to one of the regressors (in chronological order within the GLM). Thus, orthogonalization effectively controls for redundancy between the two parametric regressors within each task condition. Furthermore, in our main univariate results (Fig. 3A-E), to assess the value representations of a specific condition, we contrasted both parametric regressors in the intra-modal auditory condition against baseline. This allowed us to examine the overall variance in brain representations of value captured by both parametric regressors together, regardless of whether some component of the overall variance is shared by both regressors or assigned to the first regressor.

An alternative analysis to the one described above is to contrast individual value regressors against baseline (i.e. either *lpSV > 0* or *hpSV > 0*, see Table S5). However, we note that the estimated effects based on contrasting each of the two parametric regressors separately against the baseline have lower signal to noise ratio compared to when these regressors are both contrasted against the baseline (i.e., *lpSV + hpSV > 0*), due to the residual collinearity of regressors after orthogonalization. Therefore, only for the cross-validation results shown in Fig. 3G, this approach (i.e. either *lpSV > 0* or *hpSV > 0*) was taken, as described below.

To perform a cross-validation of our results regarding the dependence of the stimulus value representations (SVRs) in the OFC and vmPFC, we defined ROIs based on responses to the individual visual and auditory options of the inter-modal condition (*interAudVis_aSV > 0* and *interAudVis_vSV > 0*, see Table S5). For each of these contrasts, the 10 most active voxels in OFC or vmPFC were taken as an ROI, resulting on one visual and one auditory ROI in each area. We then measured the responses of each ROI to all *intra-modal* stimuli. This means that for the visual ROI, we measured responses to each individual intra-modal regressor corresponding to visual green, visual red, auditory high-pitch and auditory low-pitch stimuli (*gSV > 0, rSV > 0, hpSV > 0* and *lpSV > 0* contrasts, respectively). We then averaged the responses across all stimuli corresponding to the same modality. A similar approach was taken for the auditory ROI. This way, we ensured that the definition of ROI and the test of the effects shown in Fig. 3G were done on entirely independent data sets as inter-modal and intra-modal conditions were recorded across different runs. The effect sizes shown in Fig. 3G are the average regression coefficients (betas) from the above contrasts that were z-scored across all data of each participant. We then tested the extent to which the effect size in ROIs defined based on visual or auditory modalities exhibited selectivity for other conditions from the same modality.

On account of previous studies identifying the domain-general and domain-specific valuation areas in the vmPFC and OFC[30–32], we limited our analysis to a mask encompassing the orbital surface of frontal gyrus. This search volume (for details see Figs. S3 and S7) consisted of anatomical parcellations of orbital surface of frontal gyrus as defined in automated anatomical labelling (AAL) atlas[88,89]. Statistical maps were assessed for cluster-wise significance using a cluster-defining threshold of $t(19) = 3.58$, $P = 0.001$ for simple contrasts (see Table 1) and at $t(19) = 2.86$, $P = 0.005$ for interaction contrasts (see Tables 2 and S5); and using small volume corrected threshold of $P < 0.005$ (referred to as a small volume family-wise-error (SVFWE) correction) within the frontal search volume. Whole-brain results were inspected at FWE $p < 0.05$, and $k > 10$ (see Table 3).

## fMRI multivariate pattern classification analysis (MVPA)

For the MVPA analysis, we estimated the responses elicited by each stimulus using a GLM approach, similar to previous studies[58,59]. This GLM was identical to the univariate GLM except for two key differences: we used unsmoothed preprocessed fMRI data, and each trial of a modality condition was included in the GLM as a separate regressor modeled with stick functions at the onset time of the choice options. We did not include parametric regressors corresponding to the value of choice options in this GLM. All other regressors, including those for feedback, response, and regressors of no interest, were modeled similarly to the univariate GLM. All regressors were convolved with the canonical hemodynamic response function (HRF).

The parameter estimates (betas) from this GLM were sorted into different stimulus options (i.e., responses to red or green visual stimuli, high- or

low-pitch auditory stimuli in intra-modal conditions, and visual or auditory stimuli in inter-modal conditions) and z-scored across the entire experiment for each participant. These z-scored estimates were then fed into pattern classifiers built using LibSVM's implementation of linear support vector machines (SVMs) in MATLAB.

In our experiments, the subjective value (SV) of each stimulus was estimated from the behavioral data of individual participants using our computational framework, where SV can take any value on a continuous scale. For simplification, we binned the SV of each stimulus option into four equal bins, spanning from the lowest to the highest values of that option condition across all trials. The SVM pattern classifiers were then trained to distinguish the SV level of each option, with a chance performance of 25% (see Fig. 4).

The SVM classification followed procedures from a previous study[40]. Four types of classifiers were built: *Aud_on_Aud* and *Vis_on_Vis* (trained and tested on the same modality), and *Aud_on_Vis* and *Vis_on_Aud* (cross-modality classifiers). The classifiers were trained and tested over 100 iterations. At each iteration, half of the trials corresponding to a choice option were randomly sampled to serve as training data. *Aud_on_Aud* and *Vis_on_Vis* classifiers were then tested either on the remaining half of trials from the same choice option used for training (as shown for *Visual* and *Auditory* classifiers in Fig. 4A, B and D) or on other stimulus options from the same modality (as shown for *cross-identity* classifiers in Fig. 4A, B). For cross-modality classifiers, the classifiers were trained on one modality (with half of the trials of that modality randomly assigned to training data) and tested on a different modality (using a randomly selected half of the trials of that modality). At each iteration, classification accuracy was also measured with shuffled class labels for each classifier to estimate chance performance. The classification accuracy for all classifiers was then averaged across all iterations ($N = 100$).

We used both an ROI-based and a searchlight method. For the ROI-based method (Fig. 4A, B), the input data were the activations of all voxels within an anatomical ROI (vmPFC and OFC regions—posterior and lateral OFC—in each hemisphere from the AAL[88] atlas) in response to different stimuli from our stimulus sets: S1 = {low pitch, green, auditory} and S2 = {high pitch, red, visual}. To avoid selection bias and assess classification across the whole dataset, classifiers were built based on all possible combinations of stimuli used for training and testing. This resulted in 15 classifiers representing all binary combinations of our stimuli, S1 and S2, for training and testing. Since in a binary choice situation, the subjective value of choice options depends on each other, we excluded classifiers trained and tested on stimulus options used in the same trial (e.g., a classifier trained on intra-modal red was not tested on intra-modal green, as the value of these options depends on each other). This resulted in 12 classifiers in total, with different pairings of training and testing stimulus options used. We then averaged the classification accuracies across left and right ROIs, lateral and posterior OFC, as well as all instances of each classifier type (e.g., *Vis_on_Vis*). These averages were compared across participants against a chance level derived from classifiers trained with shuffled labels using paired t-tests.

In the searchlight method, classification was done over all voxels within a spherical searchlight (6 mm, corresponding to 54 voxels), with the classification accuracy mapped to the center voxel. This searchlight iteratively moved across all voxels of the frontal cortex (8353 voxels, see Fig. S3). This analysis used data from a subset of stimuli due to computational expense. Specifically, *Aud_on_Aud* classifiers were trained on intra-modal auditory high-pitch trials and tested on the same stimulus (intra-modal, auditory high pitch) or on auditory inter-modal stimuli. Likewise, *Aud_on_Vis* classifiers were trained on auditory high-pitch trials and tested on intra-modal green or inter-modal visual stimuli. The same logic was used for *Vis_on_Vis* and *Vis_on_Aud*, using intra-modal green stimuli for training and auditory high-pitch or inter-modal (high or low-pitch) for testing. The output images from the searchlight analysis entered a first-level analysis where contrasts between conditions were calculated. These contrast images were then smoothed (FWHM, 6 mm) and entered into a second-level analysis for statistical significance using one-sample t-tests. In Fig. 4C, the

cross-modality contrast is estimated by averaging the classification accuracies of cross-modality classifiers and comparing them against cross-modality classifiers trained with shuffled labels (testing the contrast: $(Aud\_on\_Vis + Vis\_on\_Aud) > (Aud\_on\_Vis + Vis\_on\_Aud)_{shuffled}$). Figure 4D is produced only for illustration purposes to avoid circularity of analysis, whereas statistical inferences are based on Fig. 4A, B and C.

**Effective connectivity analysis of fMRI data**

Our univariate analysis identified a number of regions, both in sensory and in frontal areas, that were modulated by the subjective value of each choice option (Fig. 3, Table 3 and Supplementary Fig. S6). We next aimed to determine how the long-range communication between these areas generates the stimulus value representations (SVRs) and the degree to which these representations are modality-specific. To this end, we investigated the effective connectivity (EC) of a network consisting of sensory and frontal regions exhibiting value modulations at the time of options' presentation by employing deterministic bilinear dynamic causal modelling (DCM) approach[60,61,90]. This approach fits a set of pre-defined patterns of EC within a model space to the fMRI time series and compares them in terms of their evidence (for details of the model space see the section under *Defining the model space for a 5-node network*).

The DCM approach requires two basic types of information for extracting time series data: the regions of interest (ROI, e.g., *audOFC*, see Fig. 5 and Regions of interest section) and the onset times of experimental conditions (e.g., intra-modal auditory condition of value task). The pre-defined model space contains the ROI information and the information on different experimental conditions can be provided by a GLM containing trial onsets of the experimental conditions that are of interest. The GLM used for effective connectivity analysis contained four unmodulated choice identity regressors. Two of these regressors represented the final choice (whether auditory or visual stimulus) for each intra-modal condition (*AudAud* or *VisVis*). The other two regressors corresponded to the inter-modal condition, separating these into auditory and visual trials based on the final choice (whether auditory or visual stimulus). Note that in intra-modal conditions (*AudAud*, *VisVis*) effective connectivity is the connectivity among sensory and frontal regions during the valuation of a particular modality condition, either auditory or visual. However, for inter-modal condition (*AudVis*), changes in EC during the valuation process occur for both auditory and visual modalities. To test the same models for their fit to both intra- and inter-modal conditions, we separated trials in the inter-modal condition according to whether the auditory or visual stimulus was selected (hence denoted by *interAud* or *interVis*, respectively). Using the same model space for both intra- and inter-modal conditions would provide insights into the underlying mechanisms that mediate valuation across different contexts, i.e., when the same or different sensory modalities are compared against each other in terms of their value. Additionally, this approach is more parsimonious than either having two separate sets of models for each condition or increasing the number or complexity of the models to account for the differences between intra- and inter-modal conditions[91].

**Regions of interest (ROIs).** ROIs for the effective connectivity analysis comprised the frontal valuation areas and the sensory regions that contained stimulus value representations for auditory and visual modalities according to the univariate analysis (Fig. 3A-E, Table 3, and Supplementary Fig. S6). The resulting five ROIs from which time series for DCM analysis were extracted were as follows: 1) The overlapping activation area for intra-modal visual and auditory value representations in vmPFC, which was obtained by performing an intersection of the two representations in vmPFC (using a logical "AND" operation in Marsbar toolbox of SPM[92]). 2) The activation area of left latOFC during intra-modal auditory condition – i.e., audOFC, 3) The activation area of left postOFC during intra-modal visual condition – i.e., visOFC, 4) bilateral activations in auditory sensory cortex – i.e., audSen, and 5) bilateral activations in visual sensory cortex – i.e, visSen. From each ROI the first

principal component of the pre-processed fMRI time series was extracted and fed to the EC analysis[93].

**Defining the model space for a 5-node network.** In order to understand how valuation is supported by a network comprising modality-general and modality-specific representations, we estimated 22 biologically plausible models for the value and the control tasks (11 models in each task). These models were developed over a base model and contained three types of connections: driving input, intrinsic, and modulatory connections. In all models, intrinsic connections were defined for every node of the network as self-connections and independent of the experimental condition (see also the Supplementary Tables S7 and S8). Because the stimuli were presented aurally or visually, two types of driving inputs to the network were defined for auditory and visual sensory cortices: 1) an input to ROI audSen in auditory and audio-visual conditions of both tasks, and 2) an input to ROI visSen in visual and audio-visual conditions of both tasks. The driving input was modelled by entering ones at the onset of stimuli options belonging to a certain condition type and zeros otherwise. The different models differed from each other with respect to the modulatory connections between nodes, which depended on the experimental conditions. The model space of all possible connectivity models would be extensive for a 5-node network[94], where a modulatory connection between any two nodes can exist in none or more of the 4 experimental conditions of a task (*intraAud*, *intraVis*, *interAud*, *interVis*) and 2 directions (directed and reciprocal). Therefore, we constrained the model space based on the following assumptions, which resulted in model 1 shown in Fig. 5A:

1. We included models with only bidirectional modulatory connections between nodes[94], based on the past findings that anatomical connectivity between two cortical areas is generally bidirectional[95]. Additionally, large connectivity databases indicate a strong likelihood of cortico-cortical connections to be bidirectional[96]. Moreover, this constraint does not imply that connection strengths would be identical for both the directed and reciprocal connections between two nodes.

2. Based on our specific hypotheses and for simplicity, we only included models that assumed two distinct and symmetric sub-networks for the valuation of stimulus options from auditory or visual modalities (auditory modality in: intraAud and interAud; visual modality in intraVis and interVis). The auditory sub-network comprised auditory valuation (vmPFC, audOFC) and sensory (audSen) ROIs and visual subnetwork comprised the visual valuation (vmPFC, visOFC) and sensory (visSen) ROIs. Models which contained modulatory connectivity between sensory areas or between OFC clusters were not included in our model space, however intrinsic connections exist between every pair of ROIs.

3. Finally, to explicitly test the plausibility of modality-specific valuation in OFC, we focused on EC patterns in which modality-specificity was either present or absent. The presence of modality-specific EC assumes that a sensory area (e.g., audSen) should only communicate to its respective value representation in OFC (i.e., audOFC) and not to others (i.e., visOFC), during experimental conditions involving that sensory modality (i.e., in intraAud, interAud). The absence of modality-specific EC assumes that additionally there are cross-modality connections (see model 1 in Fig. 5) between each sensory area and OFC representations. In this case, each sensory area (e.g., audSen) communicates not only to its respective OFC (i.e., audOFC) during the auditory conditions but also communicates to the other OFC value representations (i.e., visOFC) during all experimental conditions (i.e., intraAud, interAud, intraVis, interVis).

Based on the above assumptions, we first built a biologically plausible full model (model 1 in Fig. 5A). We then constructed the model space, exhaustively, by generating models with a subset of connections from the full model. In doing so, we took care of the following aspects: 1) we maintained the subset connections in a model symmetrical across the auditory and visual subnetworks; 2) we considered cross-modality connections either for all or for none of the conditions; and 3) we included models with each node connected to at least one other node of the network. Additionally, we also included a null model, which had no modulatory connectivity in the network for any experimental condition. This approach resulted in a connectivity model space consisting of 10 models per task (shown in Fig. 5A) plus a null model. In this model set, models 2, 6, 7, 8 assumed modality-specific EC and models 1, 3, 4, 5, 9, 10 assumed the lack of modality-specificity (i.e., the existence of cross-modality). We estimated each of the 22 models (11 in each task) individually for all 20 subjects. However, for one subject the parameter estimation did not converge and therefore, we excluded that subject from the effective connectivity analysis. Thereafter, we identified the most likely model using a group-level random effects Bayesian model selection (rfxBMS) approach[61]. The model exceedance probability used to find the best model as shown in Fig. 5B represents the probability that a particular model *m* is more likely than any other model in the model space (comprising of *M* models), given the group data. Note that the exceedance probabilities over the model space add to one[61]. Next, we estimated the connection strength parameters for connections of interest using a Bayesian parameter averaging (BPA) approach (Fig. 5C). After identifying the model that best characterized EC in the value task, we further tested the bidirectional as well as the unidirectional variants of the winning model (Fig. 6).

## Reporting summary
Further information on research design is available in the Nature Portfolio Reporting Summary linked to this article.

## Data availability
The processed source data have been deposited in the Open Science Framework (OSF) and can be accessed at https://osf.io/5vkjx/[97]. All raw datasets generated and analysed in the current study are available from the corresponding author upon request.

## Code availability
The computer codes used to conduct the experiments and analyze the data are available from the corresponding author upon request.

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

## Acknowledgements
We thank Peter Dechent and Carsten Schmidt-Samoa for their guidance and expert advice on fMRI data acquisition and experimental control during scanning. We thank Tabea Hildebrand and Jana Znaniewitz for their help with the data collection. This work was supported by an ERC Starting Grant (no: 716846) to A.P. I.K. was supported by the German Research Foundation (DFG) Collaborative Research Consortium CRC-1528 "Cognition of Interaction".

## Author contributions
S.D. and A.P. conceptualized the project and designed the main task. S.D., I.K. and A.P. designed the control task. S.D., J.E.A., and A.P. conducted the

experiments. J.E.A. assisted with the data collection and preprocessing of the data. S.D. and A.P. analysed the data. S.D. and A.P. interpreted the results and wrote the first draft of the manuscript. All authors revised the manuscript. A.P. acquired funding.

## Competing interests

The authors declare no competing interests.
