## [Peer review file · Communications Biology]

Modality-specific and modality-general representations of subjective value in frontal cortex

Corresponding Author: Dr Arezoo Pooresmaeili

Version 0:

Reviewer comments:

Reviewer #1

(Remarks to the Author)

Dang and colleagues investigate modality specificity in value encoding in the frontal cortex. In a neuroimaging study with human subjects, they present a foraging task, in which subjects are faced with a visual or an auditory symbolic representation of choice options. Furthermore, the authors introduce a complementary control task, in which subjects are not required to process value information. Implementing computational modeling and a GLM approach, they find that distinct sub-regions of the OFC encode modality-specific valuations, whereas the vmPFC – in line with the common currency approach – encodes modality-general representation of value. To further facilitate these findings, the authors conduct a functional connectivity analysis, which points towards selective interactions between sensory brain regions (auditory or visual, depending on the presentation method), and their corresponding sub-regions in the OFC.

The authors address an interesting question, introduce a very nice task design, and put forth compelling results. In particular, I truly appreciate the DCM analysis. Nevertheless, I believe the paper also suffers from a few shortcomings that need to be addressed by the authors before the manuscript can be considered fit for publication.

Major comments -

1. The design and research question presented by Dang and colleagues is very tightly related to the ones presented in Shuster & Levy (2018). The main points of difference are that in Shuster & Levy subjects face a risky-choice task rather than a foraging task, the introduction of the (very nice) control task, and that Shuster & Levy use MVPA for cross-modalities prediction of value processing in the vmPFC. Other differences, more minor ones in my opinion, include the specifics of the task-design, such that in Levy & Shuster the stimuli directly state the reward options (text representation vs. an audio recording of the lottery outcomes). In contrast, the current study uses a symbolic representation (checkered circles vs high/low pitch, respectively), which requires that the subjects associate the sensory stimuli with rewards. Both studies find that the vmPFC serves for domain-general encoding of value.

The current study's main contribution is therefore incremental, and suggests that the OFC is used for domain-specificity processing, while also pointing at the network structure between the sensory areas and the frontal cortex (via the DCM analysis).

I believe that the authors should adequately refer to Shuster & Levy already in the Introduction, and indicate that similar topics have already been explored (contrary to lines 96-98). Furthermore, in their current discussion of Shuster & Levy (lines 492-510), the authors argue that the sensory information in Shuster & Levy was redundant because subjects were not required to track past events. I am not sure I understand this argument. How tracking of past events in the current study was tied with the sensory-modality?

2. Fig. 3G does not necessarily indicate modality specificity, but rather point at significance levels. The authors should report beta weights rather than t-stats to indicate the effect-size for this analysis.

Furthermore, what were the main effects from the ANOVA analysis reported in this figure? In the visual ROI (in both panels) t-stats seem at the magnitude of ~0.8-1.2 (intra-modality trials) and ~1 (inter-modality trials), which implies non-significant activations. I am afraid that the fact that the authors document a significant interaction is solely due to the response of auditory trials in the "auditory ROI". Accordingly, the results from the inter-modality trials do not indicate that there is any modality specificity in the visual OFC. The authors should present the full statistics for this ANOVA analysis, as this is one of their main findings.

Another possibility is to follow Shuster & Levy's approach, and train a classifier on OFC activations. That is to say, if a model aimed to discriminate between high and low values trained on auditory trials, failed to predict values in the visual trials in the

“auditory ROI” (and vis-à-vis), that would be a great indication for modality-specificity in the OFC.

3. Fig. 2F is not clear to me. What does absVD mean in the control task? According to the Methods section, the feedback always states switch in 33% of trials, and so following the modelling strategy of v1 and v2 in the control task (see Methods and Fig. S3), it should be on average 0.33, which is exactly what the y-axis indicates. In other words, the y-axis in Fig. 2F does not point at some differences in subjective representation of the feedback across tasks, but rather simply shows that the switch rate was indeed 33%.

Alternatively, to show that the manipulation in the control task worked, I would use the fitted parameters from the value task to predict choices in the control task, and show that they predict on chance level.

4. The two value regressors in the GLM are anti-correlated, aren't they? This can be clearly seen from Fig. S3D-E and Table S2. The first two rows of Table S2 indicate assigned values to missed trials, but according to line 220 accuracy rates were as high as 95%, which means that anti-correlation between v1 and v2 (in the control task) was almost perfect ($r \approx -1$). Anti-correlation is also evident in the value task (but not as strong).

If true, this means that the GLM's design-matrix may suffer from multicollinearity, and that there are redundant regressors in the model. Perhaps a better modelling approach would be to model the value difference between the two options, and to add an interaction with the trial identity regressors.

Minor comments –

5. In Fig. 1, I would appreciate it if Panel B provided a better indication of how stimuli were associated with rewards. This was thoroughly explained in the Methods section, but I believe that it could help the readers to also include a graphical illustration in the figure.

6. The authors determined their sample-size ($n=20$) based on a study that used a visual-only version of their task (Serences, 2008). However, since 2008 several studies have suggested that event-related design like the current study should recruit a larger sample-size (e.g. Thirion et al., 2007, Yarkoni, 2009, and others). I do not think that the authors should recruit more subjects, especially given the fact that they had two scanning sessions per subject, which increases the statistical power of the study. The authors could address this issue in the Methods section.

7. Please provide more details about the task also in the main text. It was hard to follow without reading the Methods.

8. The feedback in the value task indicated the reward, whereas in the control task it indicated the instructions for the next trial (rather than accuracy). The authors should discuss this when detailing their GLM modelling.

9. I would appreciate some more behavioral results, as follows:

(a) Subject-specific distribution of model-fitting results. Was there heterogeneity across subjects?

(b) Have the authors tried to fit their model separately for each treatment (visual-only trials, auditory-only trials and mixed-modality trials)? Was there a difference in subjects' behavior?

10. Please provide some details on the fitting procedure of equation (7).

11. In Fig. S4, there seems to be a strong activation in the lateral ventricle. Have the authors applied grey-matter masking?

Thank you for the opportunity to review your paper and good luck!

Reviewer #2

(Remarks to the Author)

This paper examines the neural representations of subjective value across modalities (visual and auditory). 20 subjects in an fMRI experiment engaged in an instrumental-learning task. On each trial, they chose between two cues, either within modality (both visual or both auditory) or across modalities (where one cue was visual and the other auditory). The rewards associated with each cue changed dynamically throughout the experiment, such that subjects had to constantly update the value of each cue. A learning model was fit to behavior to estimate subjective values. The authors predicted overlapping representations of visual and auditory values in vmPFC, and distinct representations in OFC. They report results that are consistent with the hypothesis. They also use DCM to conduct effective connectivity analysis to select a connectivity model that best describes the relationships between sensory and value-related areas.

The paper is clearly written. The authors ask an interesting and important question, the design is quite novel, and the behavioral analysis is careful. The fMRI analysis approach, however, was not clear to me, and I'm not sure the results support the authors' conclusion – sorry if I misunderstood anything.

- In the GLM, each condition was modeled with two parametric predictors, one for the subjective value of each option. This makes perfect sense. However, the authors then contrast *both* of these predictors, together, with baseline. This seems to defeat the purpose of the design. I expected the authors to contrast each of the two predictors *separately* with baseline. This would allow straightforward testing of simple predictions: (1) in the intra-modal conditions, the two predictors should

overlap everywhere (both in the vmPFC and the OFC); (2) the two modalities should overlap in vmPFC, but not OFC; and (3) the auditory value from the inter-modal condition will overlap the one from the intra-modal condition, and same for visual values. The authors could then also validate their findings beyond the overlapping maps, by using one predictor for localizing and one for sampling the time course. In the current form, the results hinge on a slight shift in representation in the OFC, in a relatively small number of subjects. The only additional attempt at validation is the lower panel of Figure 3G, where the intra-modal conditions defined the ROIs, and the inter-modal condition was tested in these ROIs, which does not replicate the finding in the visual ROI. If I understand correctly, the top part of Figure 2G consists of double dipping (same data were used for ROI definition and for sampling) – this would be avoided by using one predictor for localization and another for sampling.

- The authors compare their findings in the value task, to a control task that is identical in sensory-motor terms, but does not involve valuation. This is an important feature of the design. However, there are three times as many trials for the value task than the control task, and the control task appears at the end of both scanning days. Isn't that a confound?

- Please provide a clearer description of the GLM with an orderly list of all the predictors and their timings in one place.

- Since the results come before the methods, a few details are a bit cryptic and should be explained (e.g. line 160-161, clarify what the sets are; line 184, LNP model).

Reviewer #3

(Remarks to the Author)

The authors use an established reward learning task to evaluate the neural basis of stimulus-value learning. They are particularly interested in understanding whether frontal brain regions represent the value of stimuli of distinct modalities separately or as part of a common code. To address this question, the authors use computational modeling to infer learned stimulus values from behavior and then use model-based fMRI to identify regions in the frontal cortex that encode subjective value of visual and auditory stimuli.

When comparing contrasts for subjective value of auditory and visual stimuli, the authors find that these contrasts substantially overlap in the ventromedial prefrontal cortex, whereas they have limited overlap in the orbitofrontal cortex. The authors conclude that these subregions of the OFC maintain modality-specific representations of value while the vmPFC maintains representations that are independent of stimulus modality. To support this claim, the authors employ dynamic causal modeling and show evidence that their data is best described by a model in which sensory regions provide input to stimulus-specific regions of OFC which then subsequently provides input to the vmPFC.

This article presents a compelling experimental framework however I have concerns about their interpretations of the data, several analysis choices, and the general clarity of the article. I believe that these concerns are addressable with additional analyses and text revisions.

Major concerns

Interpretation

a. Modality vs Identity: While these results are potentially consistent with modality-specificity, they could also be consistent with identity-specific responses. In other words, there may be distinct representations of different stimuli within a modality that are being averaged together in the current analysis. For example, previous work using multivariate methods has shown support for identity-specific representations (within stimulus modalities) in the OFC (e.g., Howard and Kahnt., 2017). Given these findings and that modality-specificity in the OFC is a major claim of this paper, the authors should address this and ideally demonstrate that their data is not more consistent with identity-specificity (i.e., are there separable representations of stimuli of the same modality?).

b. Specific vs General Representations: To conclude whether contrasts are distinct or substantially overlapping, the authors measure the Euclidean distance between the activation peaks and compare it to a threshold. This approach is potentially sensitive to preprocessing decisions (e.g., smoothing kernel size) and does not provide statistical evidence for their claim. There are alternative analyses that could be performed with the current dataset that would statistically support for their claims. For example, can distinct representations of value be detected using MVPA in the OFC? Or analogously can a vmPFC classifier trained to predict the inferred value of one stimulus modality also predict the subjective value of another stimulus modality? Additionally, since the authors are interested in showing evidence for a null hypothesis (i.e. common representation in the vmPFC), Bayes Factor mapping would be particularly informative.

Analysis

a. Model robustness: To interpret the model-based fMRI results, the authors should provide more information about the quality of the model fits for individual participants (see Serences, 2008) and across conditions. Relatedly, since they are interested in interpreting the results as evidence for subjective value learning, they should evaluate whether these parameters are recoverable and that there is not significant tradeoff.

b. DCM directionality: The evidence shown to assess the directionality is not compelling (Figure 5). The winning model is not substantially more probable than other models so these results should be interpreted with more caution and this caveat should be acknowledged by the authors. A minor comment: A model with bidirectional connections from OFC to vmPFC may conceptually contradict the claims being made here. If the vmPFC has a common code, then how does it send stimulus specific information back the appropriate sensory cortices? Are these “top-down” effects stimulus non-specific?

Clarity

a. Task design: The description and visual depiction of the task design (Figure 1) is not clear. During auditory stimulus presentation, is there an accompanying visual stimulus indicating the presence of an auditory stimulus (i.e., the speaker shown in the figure)? I imagine not, since this would be a significant confound, but it would be much clearer to directly show what participants perceptually experience during the task.

b. Modeling control task: The description of how the LNP model is adapted to model the control task (Figure 2F) is not clear. Are switches being modeled as rewarded or unrewarded trials? Does this model predict behavior in the control condition well? It is unclear whether the significant interaction shown in Figure 2F is meaningful or an artifact of a poor model fit.

c. Results by condition: Many results are shown averaged across conditions (e.g., Figure 2E) which makes it difficult to determine whether some effects could have alternative explanations. For example, are there similar filter weights across conditions? If not, the OFC results could instead be interpreted as differences in temporal integration rather than modality-specificity. The authors should demonstrate that fitted parameters are similar across conditions or provide an explanation for any differences.

Minor concerns

1. Given that signal dropout is particularly problematic in the key regions here, the authors should show evidence that this is not a concern in their dataset. For example, how does the SNR in these regions compare to other reported data sets? And/or can they clarify any steps they took to mitigate these concerns?
2. For the DCM, is the overlap across modalities in the OFC being excluded? How robust are the results if this is not considered?

Version 1:

Reviewer comments:

Reviewer #1

(Remarks to the Author)

The authors have done an excellent job addressing all three sets of reviewer comments.

I have two small notes:

- 1) It is worthwhile to report post-hoc comparisons from the ANOVA analysis reported in Fig 3G, since the interaction effect may suggest that only one of the two regions represents modality specificity. Please include the direct comparisons between SV_vis and SV_aud in the visual ROI, and respectively - SV_vis and SV_aud in the auditory ROI (OFC).
- 2) I Deeply appreciate the new MVPA analysis, and think that it improves the paper significantly. However, I am not sure how to interpret the Y-axis in Figure 4. What is the chance level? Does the axis represent the change in accuracy level or overall accuracy?

This is a nice and interesting paper that will be of interest to any scholar interested in decision-making neuroscience.

Reviewer #2

(Remarks to the Author)

The authors have done a good job responding to reviewers' comments. I do not have any further comments.

Reviewer #3

(Remarks to the Author)

The authors have addressed most of my concerns, and the paper has become much clearer and the results more convincing.

I am still having trouble understanding some of the decisions made to model the control task and conceptually what the modeled subjective value means in the control task. I originally thought this confusion was less crucial since it did not seem like the control task SV was used to model the neural signal but in their rebuttal the authors clarify that “we modeled these regressors across the entire experiment, including both value and control tasks within the same GLM”.

From clarifications made by the authors I now understand that the choices in the control task were modeled by considering

the switch trials as analogous to rewarded trials. Which means they replace r (Equation 2) with the instructive feedback in the control task. This strikes me as odd since rewarded trials are more behaviorally similar to stay trials (i.e., both increase the probability of repeating the previous trial choice). If this is the case, not predicting the choice data well would be a result of mismodeling rather than an interpretable difference in learning in the two tasks. These details aside, from a conceptual standpoint, if inferred SV in the control task is not relevant for behavior it seems strange to include it in the neural model. I would appreciate clarity on anything I may have misinterpreted here.

* The reviewers' comments are in black, our responses are in blue, and the revised sections in the main text are in green.

Point-by-point response

Reviewers' comments:

Reviewer #1 (Remarks to the Author):

Dang and colleagues investigate modality specificity in value encoding in the frontal cortex. In a neuroimaging study with human subjects, they present a foraging task, in which subjects are faced with a visual or an auditory symbolic representation of choice options. Furthermore, the authors introduce a complementary control task, in which subjects are not required to process value information. Implementing computational modeling and a GLM approach, they find that distinct sub-regions of the OFC encode modality-specific valuations, whereas the vmPFC – in line with the common currency approach – encodes modality-general representation of value. To further facilitate these findings, the authors conduct a functional connectivity analysis, which points towards selective interactions between sensory brain regions (auditory or visual, depending on the presentation method), and their corresponding sub-regions in the OFC.

The authors address an interesting question, introduce a very nice task design, and put forth compelling results. In particular, I truly appreciate the DCM analysis. Nevertheless, I believe the paper also suffers from a few shortcomings that need to be addressed by the authors before the manuscript can be considered fit for publication.

Thank you for the positive evaluation of our work and the insightful and constructive suggestions, which have significantly contributed to improving the analyses and interpretation of our results. We have thoroughly reviewed and incorporated these suggestions into our work, as outlined below.

Major comments -

1. The design and research question presented by Dang and colleagues is very tightly related to the ones presented in Shuster & Levy (2018). The main points of difference are that in Shuster & Levy subjects face a risky-choice task rather than a foraging task, the introduction of the (very nice) control task, and that Shuster & Levy use MVPA for cross-modalities prediction of value processing in the vmPFC. Other differences, more minor ones in my opinion, include the specifics of the task-design, such that in Levy & Shuster the stimuli directly state the reward options (text representation vs. an audio recording of the lottery outcomes). In contrast, the current study uses a symbolic representation (checkered circles vs high/low pitch, respectively), which requires that the subjects associate the sensory stimuli with rewards. Both studies find that the vmPFC serves for domain-general encoding of value.

The current study's main contribution is therefore incremental, and suggests that the OFC is used for domain-specificity processing, while also pointing at the network structure between the sensory areas and the frontal cortex (via the DCM analysis).

I believe that the authors should adequately refer to Shuster & Levy already in the Introduction, and indicate that similar topics have already been explored (contrary to lines 96-98). Furthermore, in their current discussion of Shuster & Levy (lines 492-510), the authors argue that the sensory information in Shuster & Levy was redundant because subjects were not required to track past events. I am not sure I understand this argument. How tracking of past events in the current study was tied with the sensory-modality?

Thank you for this thoughtful comment. We agree with the Reviewer that our work is closely related to the previous work of Shuster & Levy (2018), and we have now clarified this in the Introduction lines 94-118. We believe that their research and ours, being both conceptualised and conducted recently, have addressed an area that had been underexplored in the past: namely how the sensory modality of rewards or reward-predicting cues affects the representation of their value and whether and how the prevailing common currency framework of valuation should be revisited to accommodate the dependency of valuation on the sensory context. We however maintain that there are key differences in the design of the two studies and the methodologies as mentioned in lines 631-655. Most crucially, in the current study the expected reward was not explicitly cued in each trial, but a) it had to be linked to a specific feature of a stimulus in auditory or visual modality (pitch of tones and colour of checkerboards), and b) it had to be tracked and integrated over time, based on the outcomes of the past trials, as shown by the results of our computational model in Figure 2C. This tracking of past events must be modality-specific during inter-modal condition blocks, but our results also suggest that it remains modality-specific even during intra-modal conditions. However, we totally agree that these differences do not change the important point that the Reviewer raised: conceptually these studies are closely related, and our results can be seen as an important addition to those of Shuster and Levy (2018). To elucidate these points, we have now revised the Introduction (lines 94-118) and Discussion (see lines 612-633), copy-pasted here for your attention.

In Introduction:

“Different lines of evidence have pointed to the potential role of OFC, in particular the lateral OFC, in stimulus-specific valuation³⁰⁻³⁴, whereas vmPFC has been shown to underlie common currency coding of reward value^{7,9,11,34-40}. However, most of these insights have been derived from studies focused on visual stimuli and it remains uncertain whether they extend to the representation of the value of reward-predicting stimuli from different sensory modalities. More importantly, the neural mechanisms that generate stimulus-specific representations of value and coordinate these with common currency valuation have remained underexplored. These gaps have been noticed and addressed by a few recent studies⁴¹⁻⁴³. Specifically, Shuster and Levy⁴³ introduced an elegant way of investigating how the representation of subjective value interacts with the sensory features of stimuli through employing reward-predicting options from two different sensory modalities. Stimuli from different sensory modalities have distinct representations at the input level – for instance in the early visual and auditory cortices. This feature provides a unique opportunity to selectively trace the neural mechanisms that generate the representations of value dependent or independent of the sensory context across the brain, from the early sensory areas to the frontal valuation regions. Taking this approach, Shuster and Levy⁴³ demonstrated the existence of common representations of stimulus value in the vmPFC, whereas modality-specific representations were only observed in the sensory areas.

In the current study, we aimed to build upon these observations⁴³ and find whether modality-specific stimulus value representations (SVR) also exist in the key valuation regions in the frontal cortex. We were specifically interested in the valuation process in a dynamic foraging situation when trial-by-trial updating of computed values based on tracking modality-specific reward history is necessary. We further sought to understand *how* the putative modality-specific representation in the frontal cortex are generated through using an effective connectivity analysis approach.”

In Discussion:

“We are only aware of one neuroimaging study by Shuster and Levy⁴³ where modality-specificity of reward-predicting valuation across auditory and visual domains was examined. Using a risk-based choice task where monetary values of lotteries were either presented visually or announced aurally, this study found that the anterior portion of vmPFC represents value irrespective of the sensory modality, whereas no evidence for modality-specificity was found beyond sensory areas. Several factors could explain the absence of modality-specificity in the OFC in Shuster and Levy's study⁴³. Firstly, their design involved visual and auditory stimuli (text representation and audio recording of lottery numbers) that could be directly and unequivocally translated into explicit numeric values. This contrasts with our

study, where specific sensory features of choice options (red/green checkerboard circles and low/high pitch tones) had to be disambiguated in terms of their link to rewards. Secondly, in our design, distinct features of choice options had to be continuously tracked across trials, and their subjective value had to be updated based on choice and reward history, a key difference from previous studies⁴³. Given that one of the major roles of the OFC is updating the valuation of options based on past reward outcomes²⁴, this could explain the discrepancy between the two studies. Due to the importance of tracking sensory features of stimuli in our design, we needed to account for the covariation of sensory and reward processing¹⁸. For this, we used a control task which was similar in sensory requirements and final choice to the value task but did not require updating computed values for each sensory modality. These features of our design allowed us to uncover both modality-specific and modality-general representations of value in the frontal cortex and extend previous findings⁴³.”

2. Fig. 3G does not necessarily indicate modality specificity, but rather point at significance levels. The authors should report beta weights rather than t-stats to indicate the effect-size for this analysis.

Furthermore, what were the main effects from the ANOVA analysis reported in this figure? In the visual ROI (in both panels) t-stats seem at the magnitude of ~0.8-1.2 (intra-modality trials) and ~1 (inter-modality trials), which implies non-significant activations. I am afraid that the fact that the authors document a significant interaction is solely due to the response of auditory trials in the “auditory ROI”. Accordingly, the results from the inter-modality trials do not indicate that there is any modality specificity in the visual OFC. The authors should present the full statistics for this ANOVA analysis, as this is one of their main findings.

Another possibility is to follow Shuster & Levy’s approach, and train a classifier on OFC activations. That is to say, if a model aimed to discriminate between high and low values trained on auditory trials, failed to predict values in the visual trials in the “auditory ROI” (and vis-à-vis), that would be a great indication for modality-specificity in the OFC.

Many thanks for these observations and your helpful suggestions. We have now added two new analyses that address these concerns. One analysis amends the issues you raised with the previous univariate analysis and the second analysis adopts an MVPA approach similar to Shuster and Levi (2018) and our previous studies (Antono et al., 2023, Pooresmaeili et al., 2014^{63,64}) to examine the modality-dependence of valuation, as outlines below.

Firstly, for the univariate results reported in Figure 3G, we now take an approach recommended by Reviewer 2, avoiding the possibility of double-dipping. To achieve this, we find the ROIs shown in Figure 3G based on responses to the auditory and visual options in inter-modal condition and then test the responses of these ROIs to all other choice options in intra-modal condition. For this analysis we use betas, as you suggested, that were Z-scored across all data of an individual participants to account for differences across runs. These results are now reported in Figure 3G and 350-372 of the results section (see also lines 1074-1090 in Methods).

“To test whether the observed segregation of auditory and visual SVRs in the OFC during intra-modal conditions truly reflected a functional specialization for modality-specific valuation, we performed a cross-validation procedure (**Figure 3G**). For this, the definition of regions of interest (ROIs) and the measurement of effect were done on independent datasets to avoid double-dipping⁶⁰. Specifically, we defined the visual and auditory ROIs based on the corresponding stimulus value representations from the inter-modal condition. We then assessed the responses of these ROIs to individual parametric value regressors for each stimulus option in the intra-modal conditions (i.e., visual red, visual green, auditory high-pitch, and auditory low-pitch) and averaged the responses within each modality (see Methods, **Supplementary Information**, and **Table S5** for details).

Modality-specific valuation would require that the auditory ROI is more responsive to variations in the subjective value (SV) of auditory stimuli compared to visual stimuli, and vice versa for the visual ROI. Statistically, this would manifest as an interaction effect between ROI and the type of SV that the ROI represents. Conversely, modality-general valuation would require that auditory and visual ROIs respond similarly to both auditory and visual SVs. Examining auditory and visual ROIs in the OFC (**Figures**

3G, upper panel) and vmPFC (**Figures 3G**, lower panel) provided evidence for these types of valuation, respectively. A repeated measures ANOVA with factors: ROI (visual or auditory) in the OFC, and SV (visual or auditory), showed only an interaction between ROI and SV modality ($F[1,19] = 9.88$, $p = 0.005$), with no main effect of ROI or SV (all $ps > 0.1$). The same analysis in the vmPFC revealed no significant main or interaction effects (all $ps > 0.1$). These results further support distinct representations of subjective values for each modality in OFC, whereas in the vmPFC subjective value is represented regardless of the stimulus modality. "

Secondly, having done these clarifications, we still see the Reviewer's point that the evidence for modality-specificity based on the univariate results is not as strong as our other reported effects. To address this concern, we followed your suggestion and implemented an MVPA analysis, similar to the approach of previous studies (Shuster and Levy 2018, Antono et al 2023, Pooresmaeili et al. 2014). For this we undertook 1) an ROI-based MVPA analysis, where responses of 6 anatomical ROIs (left and right vmPFC, left and right posterior, and left and right lateral OFC, see the ROIs in Figure S7) to different combinations of visual and auditory stimuli (visual: red or green; auditory: high or low pitch and inter- or intra-modal) were used as the input to classifiers, and 2) a searchlight analysis where classification was done iteratively on all voxels with a searchlight (6mm, here we used only one condition of each modality). The searchlight analysis was done on a larger frontal ROI shown in Figure S3. The ROI-based analysis serves the purpose of showing the extent to which our classifiers trained with activations in anatomical ROIs were able to correctly determine the value of choice options when training and testing was done on the same modality as opposed to different modalities. The searchlight approach allows determining the anatomical location of regions which exhibit correct cross-classification. Doing these analyses, we could in fact confirm in both types of analysis that: 1) only in the vmPFC a classifier trained with one sensory modality was also able to classify values from the other sensory modality, hence cross-modality of valuation exists in this area confirming the findings of Shuster and Levy 2018 (see Figure 4A,B), and 2) Conversely, in OFC, cross-classification across modalities was not possible: when MVPA classifiers were trained with trials from a certain modality (say auditory), the value of the other modality could not be decoded accurately from these regions. This was not due to a general inability of the classifiers to generalize across different stimuli, as in both region, classifiers trained and tested with different stimuli from the same sensory modality (cross-identity classifiers), performed significantly above chance level. We now report these findings in the results section and depict them in the new Figure 4 pasted below for your consideration. Please note that we decided to report the MVPA results as exploratory, as this analysis was not planned initially, and we only undertook it after it was recommended by the reviewers (see also the comments of Reviewer 3). For the sake of transparency regarding our study plan, we chose to report these results as exploratory.

Therefore, we present multiple lines of converging evidence using different types of analyses to support our findings and demonstrate their robustness.

We have copy-pasted the MVPA results and the corresponding Figure below for your consideration (lines 423-471 of the Results section). See also lines 1102-1170 in the Methods and Figure S7 in the Supplementary Information.

“Modality-general and modality-specific stimulus value representations in the frontal cortex (multivariate results)

To evaluate the robustness of our findings and compare them with prior research⁴³, we conducted a multivariate pattern analysis (MVPA). This analysis is exploratory as it was not part of our initial study plan. We performed two types of MVPA, following approaches similar to previous studies^{43,63,64}. The first analysis used anatomical ROIs from the AAL atlas (lateral and posterior OFC and vmPFC; see **Figure S7**) and calculated the accuracy of MVPA classifiers built from the activation patterns of all voxels within each ROI (**Figure 4A-B**). The second analysis was a searchlight classification, where classification accuracies were iteratively calculated for all voxels in the frontal cortex (**Figure S3**). In both cases, at the first-level analysis for each participant, we computed the accuracy of classifying trials by their respective levels of subjective value. Since the subjective value estimated from our

computational model is on a continuous scale, we binned the data into four distinct levels to facilitate classification. At the group level, classification accuracies were tested against chance level (see Methods for details).

Four types of classifiers were constructed to decode the value following a previous study's approach⁴³: 1) Classifiers trained and tested on auditory trials (*Aud_on_Aud*), 2) classifiers trained and tested on visual trials (*Vis_on_Vis*), 3) classifiers trained on auditory trials and tested on visual trials (*Aud_on_Vis*), and classifiers trained on visual trials and tested on auditory trials (*Vis_on_Aud*). First, we confirmed that our ROIs (vmPFC and OFC) could accurately represent the subjective value of both visual as well as auditory stimuli by examining classifiers trained and tested on the same stimulus option (e.g., *Vis_on_Vis* trained and tested on intra-modal visual red stimuli). These are denoted as *Visual* and *Auditory* classifiers in **Figure 4A-B**. Next, we tested whether vmPFC and OFC could identify the value of different stimuli from the same sensory modality. For this, we inspected *Aud_on_Aud* and *Vis_on_Vis* classifiers trained and tested on different stimuli within the same modality. These classifiers were averaged across both modalities (*Aud_on_Aud* and *Vis_on_Vis*) to represent *cross-identity* classification accuracy. Finally, to discern whether our ROIs represent stimulus value independent of sensory modality, we examined the *cross-modality* classifiers, averaging the classification accuracies of *Vis_on_Aud* and *Aud_on_Vis* classifiers.

Using this approach, we found that in both vmPFC and OFC anatomical ROIs, subjective value was decoded significantly above chance levels for both visual and auditory options, confirming the role of these areas in representing subjective value. Additionally, both areas demonstrated above-chance cross-identity classification, indicating they could identify the value of different stimuli from the same modality (see cross-identity classifiers in **Figure 4A-B**). Importantly, only in the vmPFC did cross-modality classifiers perform significantly above chance levels. This suggests that classifiers trained with one modality could successfully discriminate the value of stimuli from another modality in the vmPFC but not in the OFC, corroborating our univariate results.

To elucidate whether modality-general representations are exclusive to the vmPFC or they exist in other regions of the frontal ROI shown in **Figure S3** and **S7**, we next examined the searchlight MVPA results (**Figure 4C**). Regions responsive to subjective value irrespective of sensory modality were defined as those exhibiting significant activations for the cross-modality contrast. Notably, this contrast showed activations exclusively in the vmPFC, corroborating the univariate results and previous findings⁴³. The lack of cross-modality activations in the OFC despite its involvement in valuation (**Figure 4A-B**) supports the idea that this region represents value in a modality-specific manner. Collectively, the univariate and multivariate results provide strong evidence for modality-specific representations in the OFC and modality-general representations in the vmPFC.”

Figure 4. Multivariate pattern analysis (MVPA) results. (A-B) MVPA results in the anatomical ROIs, defined based on the AAL atlas. Each boxplot represents the average classification accuracies for classifying the data into four levels of subjective value across all classifiers of the same type against the chance level. *Visual* and *Auditory* classifiers are a subset of *Vis-on-Vis* and *Aud-on-Aud* classifiers that were trained and tested on the same stimulus option, from either visual or auditory modality, respectively. *Cross-identity* classifiers were trained and tested on different stimuli from the same sensory modality. *Cross-modality* classifiers were trained on data from one sensory modality and tested on data from another sensory modality. (C) Searchlight MVPA results for the cross-modality contrast (*Vis-on-Aud* + *Aud-on-Vis*) shown at the peak of the depicted cluster ($P_{FWE} = 0.05$, $k = 10$, MNI coordinates: $x = 6$, $y = 52$, $z = -6$). (D) The average classification accuracy level of voxels within the cluster shown in (C) is displayed for illustration purposes. In A, B, and D, individual dots represent the data of individual participants. In A-B, *** indicates $P < 0.001$ and ** indicates $P < 0.01$ and * indicates $P < 0.05$ for a comparison against the chance level (paired t-tests) and corrected for multiple comparisons (Bonferroni correction). Chance level in all panels is calculated based on classifiers trained with shuffled class labels.

3. Fig. 2F is not clear to me. What does absVD mean in the control task? According to the Methods section, the feedback always states switch in 33% of trials, and so following the modelling strategy of v1 and v2 in the control task (see Methods and Fig. S3), it should be on average 0.33, which is exactly what the y-axis indicates. In other words, the y-axis in Fig. 2F does not point at some differences in subjective representation of the feedback across tasks, but rather simply shows that the switch rate was indeed 33%. Alternatively, to show that the manipulation in the control task worked, I would use the fitted parameters from the value task to predict choices in the control task, and show that they predict on chance level.

Thank you for this observation and for your suggestion. For the results shown in Figure 2F, we fitted the same LNP model that was used for the value task to the behavioural data of the control task. This way, we aimed to demonstrate that the LNP framework uniquely predicted learning in the value task and not in control task. For this purpose, Figure 2F shows how the difference in the subjective value (i.e., *absDV*: a representative measure of value learning) estimated from the LNP model varies as a function of the feedback provided. Based on this rationale, which is now clarified in the Results section (lines 262-287 pasted below), we decided to keep Figure 2F. However, we agree that the approach suggested by the Reviewer is indeed much more elegant and precise. Thus, now we follow your advice and implement the suggested analysis, and we report these results as well (see also lines 932-950 in Methods):

“In the control task, participants followed the instruction provided by the feedbacks with a high accuracy (i.e., $95.2\% \pm 1.33\%$ collapsed across keep/switch feedbacks), which indicated that they were aware of the task strategy. Because choices in the control task were explicitly instructed on a trial-by-trial basis, participants' decisions were not expected to reflect past reward history, unlike in the value task. This distinction was effectively captured by the LNP (**Figure 2F**): the model's estimates of how much participants updated their beliefs about options' differential value (*absDVs*) based on the feedback history showed a significant interaction ($F[1,19] = 254.7$, $p < 0.001$) between the type of the task (value or control) and the type of feedback received in the past trial (blue or yellow), see **Figure 2F**. Whereas in the value task, *absDVs* derived from the model showed a significant difference between the two types of feedbacks (mean \pm s.e.m. = 0.12 ± 0.01 and 0.74 ± 0.03 for blue and yellow feedbacks, respectively, $p < 10^{-11}$), these estimates were not significantly different in the control task (mean \pm s.e.m. = 0.27 ± 0.02 and 0.26 ± 0.02 for blue and yellow feedbacks, respectively, $p = 0.11$). This finding illustrates that the LNP model effectively captured the distinct roles of feedback in the value and control tasks.

We next examined the robustness of the model in predicting choices in the two tasks, using a leave-one-out procedure described in a previous study⁵⁴. For this, we tested parameter τ (**Figure 2C**, see also equation 3) and the sigmoidal choice probability function (solid magenta line as shown in **Figure 2D**, and equation 7). This leave-one-out procedure was iterated for all 18 blocks of the value task and the mean prediction accuracy across participants was $74.31\% (\pm 1.17 \text{ s.e.m.})$, being significantly higher than the chance level (paired-sample t-test results when compared with chance level accuracy (50%): $t[19] = 20.69$, $p < 0.001$). In contrast, when the fit parameters were estimated from the value task and tested on the control task, participants' choices were only predicted at chance level $49.12\% (\pm 1.14 \text{ s.e.m.})$ ($t[19] = -0.77$, $p = 0.4504$). These results support our notion that the LNP framework uniquely predicted learning and choices in the value task but not in the control task.”

4. The two value regressors in the GLM are anti-correlated, aren't they? This can be clearly seen from Fig. S3D-E and Table S2. The first two rows of Table S2 indicate assigned values to missed trials, but according to line 220 accuracy rates were as high as 95%, which means that anti-correlation between v_1 and v_2 (in the control task) was almost perfect ($r \cong -1$). Anti-correlation is also evident in the value task (but not as strong). If true, this means that the GLM's design-matrix may suffer from multicollinearity, and that there are redundant regressors in the model. Perhaps a better modelling approach would be to model the value difference between the two options, and to add an interaction with the trial identity regressors.

Thanks for pointing this out. As mentioned in the results section on the fMRI univariate analysis, our primary goal was to determine the degree to which the subjective values (SVs) assigned to each option in a binary choice situation are dependent or independent of their sensory modality. Therefore, for us it was crucial to measure the activations induced by the SVs of either of the two choice options rather than activations induced by their differential values. Given this, we believe that the current GLM is the appropriate choice for testing our specific hypothesis, as clarified in lines 1030-1050.

Moreover, regarding redundancy/collinearity between parametric regressors, we have now clarified how this issue was mitigated in lines 1051-1066 of the Methods section pasted below for your attention. “Note that our GLMs included two parametric regressors each modeling one of the two choice options in a trial of value and control tasks. This inherently introduces some degree of (anti)correlation between

the two parametric regressors, as in every binary choice situation the subjective value of one option is influenced to some extent by the subjective value of the other option. To mitigate this correlation, we employed two strategies. Firstly, we modeled these regressors across the entire experiment, including both value and control tasks within the same GLM. This reduces collinearity, as during intervals when participants perform a different task, the two parametric regressors are not correlated. Secondly, we utilized orthogonalization of regressors based on the standard method implemented in SPM¹⁰¹. This technique removes shared variance between the regressors due to collinearity and assigns it to one of the regressors (in chronological order within the GLM). Thus, orthogonalization effectively controls for redundancy between the two parametric regressors within each task condition. Furthermore, in our main univariate results (**Figure 3A-E**), to assess the value representations of a specific condition, we contrasted both parametric regressors in the intra-modal auditory condition against baseline. This allowed us to examine the overall variance in brain representations of value captured by both parametric regressors together, regardless of whether some component of the overall variance is shared by both regressors or assigned to the first regressor.”

Lastly, to fully address this concern and provide a comparison to the approach you suggested (modelling the value difference between two options), we now report the results of a GLM with a single parametric regressor per trial, modelling the absolute differential value of the two options (see Supplementary Figure S9 and Supplementary Text). This analysis closely reproduces our main results based on modelling two parametric regressors and additionally indicates that modality-specificity in the OFC is maintained not only during valuation but also during the final choice. Furthermore, the new MVPA results, which closely reproduce the univariate results, were based on an analysis where each trial was modelled by a single event-related stick regressor at the onset time of choice options. Given the similarities in our findings using these different approaches, we are confident that our results were not affected by the specific modelling of the GLMs employed.

Minor comments –

5. In Fig. 1, I would appreciate it if Panel B provided a better indication of how stimuli were associated with rewards. This was thoroughly explained in the Methods section, but I believe that it could help the readers to also include a graphical illustration in the figure.

Thanks for this suggestion. We have now changed Figure 1 and illustrate how stimuli were associated with rewards (cf. Figure 1 and its legends lines 150-175).

6. The authors determined their sample-size (n=20) based on a study that used a visual-only version of their task (Serences, 2008). However, since 2008 several studies have suggested that event-related design like the current study should recruit a larger sample-size (e.g. Thirion et al., 2007, Yarkoni, 2009, and others). I do not think that the authors should recruit more subjects, especially given the fact that they had two scanning sessions per subject, which increases the statistical power of the study. The authors could address this issue in the Methods section.

Thanks for this suggestion. We now mention this issue in lines 735-738 of the Methods section.

“Furthermore, to ensure adequate statistical power for detecting valuation across different sensory modality configurations, each participant underwent scanning during two sessions, thereby providing a sufficient number of repetitions for each configuration (see under Experimental Design).”

7. Please provide more details about the task also in the main text. It was hard to follow without reading the Methods.

Thank you for this suggestion. We have thoroughly revised the main text including experimental details crucial for understanding the procedures in the main text, including but not limited to the caption of figures (cf. Figure 1) and lines 186-203 of the Results.

8. The feedback in the value task indicated the reward, whereas in the control task it indicated the instructions for the next trial (rather than accuracy). The authors should discuss this when detailing their GLM modelling.

Thanks for raising this point. We now clarify this feature in lines 1006-1010:

“Additionally, we also modelled the response to the feedback displays in each trial. Please note that in the value task, the feedback display in a trial indicated whether the chosen option in the current trial was rewarded or not; while in the control task, the feedback indicated whether to switch or keep the current trial’s choice in the next trial (For more details, refer to the **Experimental Design** section).”

9. I would appreciate some more behavioral results, as follows:

(a) Subject-specific distribution of model-fitting results. Was there heterogeneity across subjects?

Thanks for this suggestion. We now show the data of individual subjects while showing the model-fitting results in Figure 2A, B and E and supplementary figures depicting the model fits to individual sensory modalities **Figure S2**, see the response to your point b below. This allows for a clearer representation of the spread of effects across participants.

(b) Have the authors tried to fit their model separately for each treatment (visual-only trials, auditory-only trials and mixed-modality trials)? Was there a difference in subjects’ behavior?

Yes, we indeed conducted the model-fitting for individual conditions (auditory, visual, audio-visual) of the value task and we did not find any significant difference in parameters of the best-fitting curves across the sensory modalities (see lines 239-242 of the Results and Figure S2 in the Supplementary Information). We therefore presented the behavioural data and the model-fitting results in Figure 2 of the manuscript by collapsing all conditions together for the value task and did not include the results for individual conditions in the main text because the choice patterns were found consistent across the sensory modalities. These results are now reported in Figure S2 and Table S3 of the Supplementary Information. Figure S2 is pasted under responses to Reviewer 3 for your consideration.

10. Please provide some details on the fitting procedure of equation (7).

Thanks for your suggestion. We have now added more details on the fitting procedure of equation (7) in lines 932-939 of the Methods section.

“To assess the *decision-making phase*, i.e. the effect of the differential value on choices, we separated trials of the value task into 20 bins of equal size, based on their differential value $dv(t)$ (see equation (5)). The probability of choosing option S_1 in a bin was obtained by computing the proportion of S_1 choices made in the subset of binned trials. Intuitively, as the differential value between the stimuli options increases, the probability of choosing S_1 should also increase (see scatter plot in Figure 2D). Thereafter, the probability of choosing option S_1 was approximated by fitting a normal cumulative distribution function (equation (7)).

11. In Fig. S4, there seems to be a strong activation in the lateral ventricle. Have the authors applied grey-matter masking? In all figures, we have overlaid the statistical maps on canonical single subject T1 structural images, without using any grey-matter masking.

Thank you for the opportunity to review your paper and good luck!

Thank you so much for your extremely helpful feedback!

Reviewer #2 (Remarks to the Author):

This paper examines the neural representations of subjective value across modalities (visual and auditory). 20 subjects in an fMRI experiment engaged in an instrumental-learning task. On each trial, they chose between two cues, either within modality (both visual or both auditory) or across modalities (where one cue was visual and the other auditory). The rewards associated with each cue changed dynamically throughout the experiment, such that subjects had to constantly update the value of each cue. A learning model was fit to behavior to estimate subjective values. The authors predicted overlapping representations of visual and auditory values in vmPFC, and distinct representations in OFC. They report results that are consistent with the hypothesis. They also use DCM to conduct effective connectivity analysis to select a connectivity model that best describes the relationships between sensory and value-related areas.

The paper is clearly written. The authors ask an interesting and important question, the design is quite novel, and the behavioral analysis is careful. The fMRI analysis approach, however, was not clear to me, and I'm not sure the results support the authors' conclusion – sorry if I misunderstood anything.

1(a) In the GLM, each condition was modeled with two parametric predictors, one for the subjective value of each option. This makes perfect sense. However, the authors then contrast **both** of these predictors, together, with baseline. This seems to defeat the purpose of the design. I expected the authors to contrast each of the two predictors **separately** with baseline. This would allow straightforward testing of simple predictions: (1) in the intra-modal conditions, the two predictors should overlap everywhere (both in the vmPFC and the OFC); (2) the two modalities should overlap in vmPFC, but not OFC; and (3) the auditory value from the inter-modal condition will overlap the one from the intra-modal condition, and same for visual values.

Thank you for this careful observation and for your suggestion. This concern was addressed in two ways: 1) implementing the analysis that you suggested, 2) a whole new set of multivariate pattern classification analyses now shown in the new Figure 4. Below, we explain these after each other.

First approach: As you noticed, we did contrast **both** parametric value regressors (or predictors) of a condition together against the baseline because we were primarily interested in determining the separation between auditory and visual value representations in OFC and vmPFC, and not in individual representations of a particular type of stimulus, say, for low pitch or high pitch in intra-modal auditory condition or red or green colours in intra-modal visual condition. However, the alternative analysis and its expected outcome, as suggested by the Reviewer are interesting and insightful. Thus, in addition to our main analysis where we combine the two value regressors, we performed the suggested analysis by the Reviewer and report it as additional supporting evidence (see lines 346-349 of the main text and lines 405-449 of the Supplementary text and **Table S5**).

Specifically, we contrasted *each* of the value regressors from the individual conditions (AudAud, VisVis, AudVis) of the value task *separately* against the baseline, as you suggested. We also used the estimated responses based on this method to calculate the results shown on Figure 3G. Importantly, we observed largely similar results when either **both** regressors or **each regressor separately** were inspected, attesting to the robustness of our findings: a) the activations shown in Supplementary Table S5 are in similar locations compared to the activations shown in Figure 3 and Table 1, and b) the results shown in Figure 3G corroborate our main results (see our response to point 1b).

However, we decided to keep the main analyses based on contrasting both parametric value regressors of a modality configuration against the baseline. The rationale behind this is that modelling both regressors together is less affected by the correlation between regressors, thus providing a better estimation of responses elicited by the subjective value of a specific modality condition (e.g., intra-modal visual or auditory). The reason for this is that in each trial, there was some degree of (anti)correlation between the two parametric regressors, as is typical in binary choice situations: the subjective value of one option depends, to some extent, on the subjective value of the other option.

Through the orthogonalization process implemented in SPM, this shared variance is assigned to the previous regressor as regressors are entered into the GLM.

By contrasting both parametric regressors together against the baseline, we reduce the effect of residual collinearity. This approach allows us to examine the overall variance captured by both parametric regressors together, regardless of whether some component of the overall variance is shared by both regressors or assigned to the first regressor. These details are outlined in Methods, lines 1050-1091.

Second approach: We now report a whole set of new MVPA analysis in which classification was done separately for every stimulus in our sets (See Figure 4A-B pasted above, lines 423-471 of the Results section and lines 1102-1170 in the Methods). We found largely similar results based on these analyses.

Based on the reasons outlined above, we do report the results of the suggested analysis in Figure 3G and in Supplementary Tables S5, but we kept the rest of analyses, e.g. DCM, and our inferences based on the results obtained by modelling both regressors throughout.

1(b) The authors could then also validate their findings beyond the overlapping maps, by using one predictor for localizing and one for sampling the time course. In the current form, the results hinge on a slight shift in representation in the OFC, in a relatively small number of subjects. The only additional attempt at validation is the lower panel of Figure 3G, where the intra-modal conditions defined the ROIs, and the inter-modal condition was tested in these ROIs, which does not replicate the finding in the visual ROI. If I understand correctly, the top part of Figure 2G consists of double dipping (same data were used for ROI definition and for sampling) – this would be avoided by using one predictor for localization and another for sampling.

In light of the Reviewer’s concern about datasets used for localisation and hypothesis testing and acknowledging the importance of avoiding double-dipping⁶⁰, we undertook the following analyses (as also pointed out in response to 1(a)):

- 1) Firstly, to avoid the possibility of double-dipping, we now find the functional ROIs based on responses to inter-modal condition, recorded in a separate run and different from the clusters shown in Figure 3A-E. Specifically, we use the responses to visual *or* auditory inter-modal stimuli to define the visual and auditory ROIs shown in Figure 3G and then test the responses of these ROIs to red or green visual intra-modal and high or low pitch intramodal auditory cases. These results are now reported in Figure 3G and 350-372 of the results section (see also lines 1074-1091 in Methods).

“To test whether the observed segregation of auditory and visual SVRs in the OFC during intra-modal conditions truly reflected a functional specialization for modality-specific valuation, we performed a cross-validation procedure (**Figure 3G**). For this, the definition of regions of interest (ROIs) and the measurement of effect were done on independent datasets to avoid double-dipping⁶⁰. Specifically, we defined the visual and auditory ROIs based on the corresponding stimulus value representations from the inter-modal condition. We then assessed the responses of these ROIs to individual parametric value regressors for each stimulus option in the intra-modal conditions (i.e., visual red, visual green, auditory high-pitch, and auditory low-pitch) and averaged the responses within each modality (see Methods, **Supplementary Information**, and **Table S5** for details).

Modality-specific valuation would require that the auditory ROI is more responsive to variations in the subjective value (SV) of auditory stimuli compared to visual stimuli, and vice versa for the visual ROI. Statistically, this would manifest as an interaction effect between ROI and the type of SV that the ROI represents. Conversely, modality-general valuation would require that auditory and visual ROIs respond similarly to both auditory and visual SVs. Examining auditory and visual ROIs in the OFC (**Figures 3G**, upper panel) and vmPFC (**Figures 3G**, lower panel) provided evidence for these types of valuation, respectively. A repeated measures ANOVA with factors: ROI (visual or auditory) in the OFC, and SV (visual or auditory), showed only an interaction between ROI and SV

modality ($F[1,19] = 9.88, p = 0.005$), with no main effect of ROI or SV (all p s > 0.1). The same analysis in the vmPFC revealed no significant main or interaction effects (all p s > 0.1). These results further support distinct representations of subjective values for each modality in OFC, whereas in the vmPFC subjective value is represented regardless of the stimulus modality. "

- 2) Additionally, we also undertook a whole new set of MVPA analyses providing robust evidence for our findings beyond the evidence shown in Figure 3G. These analyses (reported in 423-471 of the Results section and lines 1101-1170 of Methods) explicitly use classifiers trained on the data of each individual option in our stimulus sets (for instance Visual: Red or Green; Auditory: High or Low pitch), similar to the logic that the reviewer suggested. In our MVPA analyses, we took great care that no circular analysis is done: 1) we used anatomical ROIs from an atlas (AAL atlas) to build classifiers shown in Figure 4A-B). These classifiers were trained on every stimulus in our sets and tested on every other stimulus from the different modality, thus avoiding any selection bias (see also lines 1102-1170). Lastly, we also confirm our results in a searchlight analysis that is completely independent of any voxel selection. The converging results from all these analyses corroborates the robustness of our findings.

2 - The authors compare their findings in the value task, to a control task that is identical in sensory-motor terms, but does not involve valuation. This is an important feature of the design. However, there are three times as many trials for the value task than the control task, and the control task appears at the end of both scanning days. Isn't that a confound?

We agree with the reviewer that the number of trials in the control task is 1/3 of trials in the value task. However, we note that: 1) the value task included 3 reward ratios over the options (S1:S2 = 1:3, 1:1, and 3:1), whereas this difference between options did not exist for the control task, 2) the control task involved a very simple cognitive operation: to follow or not follow an instruction compared to the foraging task involved in the value task and consequently the associated responses to stimuli are expected to be less variable. Based on these results and since we already had two very long scanning sessions for all conditions of the value task, we reasoned that the 1/3 of trials in control task will be enough to capture the most essential responses, i.e. choice related responses, elicited by this task. 3) Importantly, we do show that this task was powerful enough to serve its intended purpose, which was to reflect sensory responses elicited by different conditions that were independent of stimulus value, and that we were able to identify activations related to the final choice in the vmPFC with this task (see Table 2, Figure S5 and Table S6 in the Supplementary Information, as well as the explanations in line 375-393 of the main text, which are pasted below for your consideration).

"An alternative explanation for the segregation in modality-wise representations in the OFC is that rather than reflecting the functional specialization of neural responses for the visual and auditory value processing, they reflected differences in the sensory properties of stimuli. To rule out this possibility, we next examined the control task (see **Table 2** and **Table S4**, **Table S6** and **Figure S5**). Crucially, the modality-specific activations in OFC were absent in the control task when the same contrasts as in the value task were examined, demonstrating that OFC representations reflect the trial-by-trial updating of stimulus-value associations rather than the sensory features of stimuli or simple choice based on the instruction. On the contrary, in the control task activations overlapping with the modality-general representations in vmPFC were found (**Figure S5**). Inspecting contrasts which explicitly tested the value and the control task against each other supported these results (**Table 2**). Contrasting the unmodulated trial identity regressors which modelled the responses to different sensory modalities revealed no significant difference between the two tasks, confirming that the control task was powerful enough to capture the sensory responses to the stimuli. Contrasting the parametric regressors however showed strong bilateral activations in postOFC in the value task, supporting the involvement of the OFC in representing the value of stimuli beyond their sensory properties. The lack of activations in the vmPFC for this contrast further highlights a general role of this area in representing the final choice irrespective of whether or not choices were informed by value or were instructed."

Based on these reasons, we believe that the control task had sufficient power to achieve its intended objectives.

3 – Please provide a clearer description of the GLM with an orderly list of all the predictors and their timings in one place.

Thanks for pointing out the need for clarifications regarding the GLMs. To address this concern, we have tried to improve our description of the GLMs in the text (see for instance lines: 308-313), and additionally added a table (Table S3) illustrating the definition of all regressors to the Supplementary Information.

4 - Since the results come before the methods, a few details are a bit cryptic and should be explained (e.g. line 160-161, clarify what the sets are; line 184, LNP model).

This is a great suggestion. We have now improved the text in the above-mentioned lines and elsewhere to increase the clarity of the text.

Reviewer #3 (Remarks to the Author):

The authors use an established reward learning task to evaluate the neural basis of stimulus-value learning. They are particularly interested in understanding whether frontal brain regions represent the value of stimuli of distinct modalities separately or as part of a common code. To address this question, the authors use computational modeling to infer learned stimulus values from behavior and then use model-based fMRI to identify regions in the frontal cortex that encode subjective value of visual and auditory stimuli.

When comparing contrasts for subjective value of auditory and visual stimuli, the authors find that these contrasts substantially overlap in the ventromedial prefrontal cortex, whereas they have limited overlap in the orbitofrontal cortex. The authors conclude that these subregions of the OFC maintain modality-specific representations of value while the vmPFC maintains representations that are independent of stimulus modality. To support this claim, the authors employ dynamic causal modeling and show evidence that their data is best described by a model in which sensory regions provide input to stimulus-specific regions of OFC which then subsequently provides input to the vmPFC.

This article presents a compelling experimental framework however I have concerns about their interpretations of the data, several analysis choices, and the general clarity of the article. I believe that these concerns are addressable with additional analyses and text revisions.

We thank the reviewer for their positive evaluation of our work and for the very insightful and constructive suggestions, which have substantially improved the quality and clarity of our manuscript.

Major concerns

Interpretation

a. Modality vs Identity: While these results are potentially consistent with modality-specificity, they could also be consistent with identity-specific responses. In other words, there may be distinct representations of different stimuli within a modality that are being averaged together in the current analysis. For example, previous work using multivariate methods has shown support for identity-specific representations (within stimulus modalities) in the OFC (e.g., Howard and Kahnt., 2017). Given these findings and that modality-specificity in the OFC is a major claim of this paper, the authors should address this and ideally demonstrate that their data is not more consistent with identity-specificity (i.e., are there separable representations of stimuli of the same modality?)

Thank you for this comment. We acknowledge that modality-specificity and identity-specificity share conceptual overlaps. As outlined in the introduction, stimulus-specific or identity-specific valuation is a broader term that encompasses any reliance of valuation on specific stimulus features. Within this context, modality-specificity can be considered a subset of identity-specificity, not directly derived from a common currency model of valuation. However, we propose that the segregation of value representations across sensory modalities is more pronounced than the segregation within sensory modalities due to the greater featural distance between modalities compared to within modalities. This conjecture is based on the following reasons.

- 1) In light of your comment and that of Reviewer 2, we have now added the Supplementary Table S5, showing the value representations of different stimuli with a sensory modality (i.e. responses elicited by each parametric value regressor). For instance, in intra-modal visual trials, we show the stimulus value (SV) representations separately for red and green stimuli, and similarly for auditory stimuli (here we have high and low pitch features). We find that unlike the modality-specific representations, the SVRs within the same modality are overlapping or much closer in distance. This means that the representation of SV for green and red stimuli in intra-modal trials are adjacent to each other and likewise for the auditory stimuli.
- 2) Additionally, we used the activations shown in Table S5 to cross-validate the reported modality-specificity, as shown in the new Figure 3G. Here, we define visual or auditory ROIs based on the SVs of one stimulus in the same modality (e.g. visual ROIs are defined based on visual inter-modal condition), but effects are tested based on responses of these ROIs to other stimulus options in the same modality, but in different conditions (e.g. responses to red or green stimuli of the intra-modal conditions). This analysis reproduced the modality-specificity effect and further shows that modality-specificity can be observed across all stimulus features of the visual or auditory modalities.
- 3) In our exploratory MVPA analysis, we explicitly compare cross-identity and cross-modality classification of subjective value. The cross-identity classifiers were shown to be able to classify different stimuli from the same sensory modality in both vmPFC and OFC (see Figure 4A-B, also pated in the response letter), whereas cross-modality classifiers (which could generalize across modalities) only had a above-chance accuracy in vmPFC.

Collectively, these results suggest that the segregation of neural function for coding the stimulus value of different sensory modalities is larger than the segregation for representing the value within the same sensory modality. However, we think that this intriguing question requires further investigations in the future that are beyond the scope of the current study. We, therefore, suffice to present the above results and discuss them in the context of the main aim of the study, i.e. revealing the dependence of valuation on sensory modality (see also the response to your point b below).

b. Specific vs General Representations: To conclude whether contrasts are distinct or substantially overlapping, the authors measure the Euclidean distance between the activation peaks and compare it to a threshold. This approach is potentially sensitive to preprocessing decisions (e.g., smoothing kernel size) and does not provide statistical evidence for their claim. There are alternative analyses that could be performed with the current dataset that would statistically support for the authors claims. For example, can distinct representations of value be detected using MVPA in the OFC? Or analogously can a vmPFC classifier trained to predict the inferred value of one stimulus modality also predict the subjective value of another stimulus modality? Additionally, since the authors are interested in showing evidence for a null hypothesis (i.e. common representation in the vmPFC), Bayes Factor mapping would be particularly informative.

Many thanks for these careful observations and helpful suggestions. We now report the results of a whole new set of MVPA analyses which exactly follow your recommendations (see lines 423-471 of the Results section, lines 1101-1170 in Methods and Figure 4, copy-pasted above in response to Reviewer 1). Through these analyses we show 1) there is a common representation of value in vmPFC as classifiers trained to decode stimulus value (SV) in one sensory modality can also decode SV in another modality, 2) Crucially, these common representations are different from value representations

in OFC where classifiers trained with the data from one modality cannot decode SV in another modality (no cross-classification). These additional analyses provide compelling evidence for the central claim of the study and although they are largely in line with our univariate results, they do not suffer from the confounding issues that you rightfully pointed out. However, since the MVPA analysis was not part of our original study plan and the results of MVPA are largely conforming them, we still base the subsequent analyses, i.e. DCM on univariate results.

Analysis

a. Model robustness: To interpret the model-based fMRI results, the authors should provide more information about the quality of the model fits for individual participants (see Serences, 2008) and across conditions. Relatedly, since they are interested in interpreting the results as evidence for subjective value learning, they should evaluate whether these parameters are recoverable and that there is not significant tradeoff.

Thanks for this suggestion. We are now providing more information about the quality of model fits in Results section, lines 277-287 and Supplementary Information (see also the Discussion: lines 574-580). 1) Following the procedure done in Serences 2008, we show that in the value-based task the LNP model provides a good fit to the data, whereas in the control task it does not. 2) we show the model fitting results for each modality, demonstrating that there was no difference in the quality of fits across modalities (see the new Figure S2 and the response to your point c below). 3) The SV representations are directly derived from the modulations induced by parametric regressors which represent the model fits. If we understand the question correctly, one way to recover the model parameters is to test the correlation of fMRI activations with behavioural fits. While this can be easily done, we think this would be circular and unnecessary in our case since the fMRI activations we show in Figure 3 do in fact represent the model fits derived from behavioural data.

b. DCM directionality: The evidence shown to assess the directionality is not compelling (Figure 5). The winning model is not substantially more probable than other models so these results should be interpreted with more caution and this caveat should be acknowledged by the authors. A minor comment: A model with bidirectional connections from OFC to vmPFC may conceptually contradict the claims being made here. If the vmPFC has a common code, then how does it send stimulus specific information back the appropriate sensory cortices? Are these “top-down” effects stimulus non-specific?

Thanks for pointing this out. In fact, the former Figure 5 (currently Figure 6) compares different versions of the winning model derived from the model space shown in Figure 5. So, we first find the winning model in comparison to all possible connectivity schemes of the network (comprising vmPFC, OFC and sensory areas) and we then test whether bidirectional or unidirectional connections between different areas, especially the OFCs and sensory areas, prevail in this model (Model 6 in Figure 5). Therefore, it is expected that different versions of the winning model have very high probability. It is also noteworthy that model 1 and 2, both containing feedback connections between OFC nodes and sensory areas outperform all other models. However, we completely agree with the Reviewer that these results should not be overinterpreted and we now point out their limitations in lines 685-691 of the Discussion and pasted below.

“However, it is crucial to interpret the effective connectivity results with caution. For example, as illustrated in Figure 6, various versions of the winning model, differing in their mono- versus bi-directional connectivity, produced similarly good fits to the data. Therefore, drawing definitive conclusions about the specific modes and directions of connectivity across the networks shown in Figures 5 and 6 remains challenging at this stage. Nevertheless, our study provides a valuable framework for exploring how the representation of value and sensory features across the brain underpins valuation.”

Regarding the bidirectional connections between the vmPFC and OFC, in fact this may reflect other relay stations between the two regions which disambiguate the modality of the value representations or

may be due to the insufficient power of our data to detect differences at the level of vmPFC. We have now briefly pointed out this in the Discussion lines 601-608.

Clarity

a. Task design: The description and visual depiction of the task design (Figure 1) is not clear. During auditory stimulus presentation, is there an accompanying visual stimulus indicating the presence of an auditory stimulus (i.e., the speaker shown in the figure)? I imagine not, since this would be a significant confound, but it would be much clearer to directly show what participants perceptually experience during the task.

Thanks for this suggestion. Indeed, the speaker was not shown. In light of Reviewers' comments, we have now amended this figure extensively. Please see the new Figure 1 and its caption.

b. Modeling control task: The description of how the LNP model is adapted to model the control task (Figure 2F) is not clear. Are switches being modeled as rewarded or unrewarded trials? Does this model predict behavior in the control condition well? It is unclear whether the significant interaction shown in Figure 2F is meaningful or an artifact of a poor model fit.

Thank you for these observations regarding modelling of control task and figure 2F. We agree with the Reviewer that the explanations regarding the comparison of the control and value task in general and modelling results shown in Figure 2F in particular were unclear. We have now extensively revised these sections, which we copy below for your consideration.

In lines 834-838 of the Methods section, we clarify the switch rate:

“Similar to the reward structure in the value task, switches were assigned independently and stochastically to the options in an equiprobable manner with an average switch rate of 33%. Thus, on any trial when a participant earned a switch from a chosen option, yellow feedback was displayed indicating that they should switch their choice to the other option on the next trial.”

Therefore, indeed as the reviewer noted, switches in control task were being modelled as “rewarded trials” in the value task.

In lines 262-281 of the Results section and 932-950 of the Methods, we clarify the modelling results shown in Figure 2F and also evaluate the quality of model fits to both tasks, copy-pasted here for your attention:

“In the control task, participants followed the instruction provided by the feedbacks with a high accuracy (i.e., $95.2\% \pm 1.33\%$ collapsed across keep/switch feedbacks), which indicated that they were aware of the task strategy. Because choices in the control task were explicitly instructed on a trial-by-trial basis, participants' decisions were not expected to reflect past reward history, unlike in the value task. This distinction was effectively captured by the LNP (**Figure 2F**): the model's estimates of how much participants updated their beliefs about options' differential value (*absDVs*) based on the feedback history showed a significant interaction ($F[1,19] = 254.7$, $p < 0.001$) between the type of the task (value or control) and the type of feedback received in the past trial (blue or yellow), see **Figure 2F**. Whereas in the value task, *absDVs* derived from the model showed a significant difference between the two types of feedbacks (mean \pm s.e.m. = 0.12 ± 0.01 and 0.74 ± 0.03 for blue and yellow feedbacks, respectively, $p < 10^{-11}$), these estimates were not significantly different in the control task (mean \pm s.e.m. = 0.27 ± 0.02 and 0.26 ± 0.02 for blue and yellow feedbacks, respectively, $p = 0.11$). This finding illustrates that the LNP model effectively captured the distinct roles of feedback in the value and control tasks.

We next examined the robustness of the model in predicting choices in the two tasks, using a leave-one-out procedure described in a previous study⁵⁴. For this, we tested parameter τ (**Figure 2C**, see also equation 3) and the sigmoidal choice probability function (solid magenta line as shown in **Figure 2D**, and equation 7). This leave-one-out procedure was iterated for all 18 blocks of the value task and the

mean prediction accuracy across participants was 74.31% (± 1.17 s.e.m.), being significantly higher than the chance level (paired-sample t-test results when compared with chance level accuracy (50%): $t[19] = 20.69$, $p < 0.001$). In contrast, when the fit parameters were estimated from the value task and tested on the control task, participants' choices were only predicted at chance level 49.12% (± 1.14 s.e.m.) ($t[19] = -0.77$, $p = 0.4504$). These results support our notion that the LNP framework uniquely predicted learning and choices in the value task but not in the control task.”

c. Results by condition: Many results are shown averaged across conditions (e.g., Figure 2E) which makes it difficult to determine whether some effects could have alternative explanations. For example, are there similar filter weights across conditions? If not, the OFC results could instead be interpreted as differences in temporal integration rather than modality-specificity. The authors should demonstrate that fitted parameters are similar across conditions or provide an explanation for any differences.

Thank you for these observations and for your suggestions. We now show the model fits per condition in the Supplementary Information (Figure S2 and pasted below) and explicitly measure whether the model fits/parameters differ between conditions. Based on these analyses and the analysis of RT and choice data detailed in the Supplementary Information, we can confidently rule out that the differences in OFC representations of modality-wise values are due to differences in temporal features or behavioural characteristics or physical characteristics of the auditory and visual stimuli.

Figure S2: Mean of model-fit parameters across participants for the individual sensory modality conditions of the value task - (A) Auditory, (B) Visual, and (C) AudioVisual. Each dot represents the data of an individual participant. In order to test whether significant differences existed across sensory modality conditions for individual model-fit parameters, we performed a one-way repeated-measures ANOVA. We found no main effect of sensory modality on any of the three parameters: Tau ($F[2,38] = 1.72$, $p = 0.19$), Mu ($F[2,38] = 2.03$, $p = 0.15$), Sigma ($F[2,38] = 0.75$, $p = 0.48$). Therefore, the results shown in Figure 2E were pooled across all conditions.

Minor concerns

1. Given that signal dropout is particularly problematic in the key regions here, the authors should show evidence that this is not a concern in their dataset. For example, how does the SNR in these regions compare to other reported data sets? And/or can they clarify any steps they took to mitigate these concerns?

Thanks for pointing this out. We have now added this information to the Methods (lines 962-969) and Supplementary Information. To account for the signal dropout in the frontal regions the following steps were taken during the data acquisition: 1) We used a data acquisition protocol that minimizes signal dropout in the regions⁴: for this a PA phase encoding direction was used in combination with a rather high in-plane spatial resolution and rather thin slices (2x2x2mm) for our EPIS, and 2) we took special steps during the data acquisition: after every fMRI run we acquired a field map (instead of just one field map per session) to have an as close as possible match between fMRI and field map to correct image distortions during the data analysis.

To explicitly test the possibility of major signal drop-out in frontal areas, we also report the results of an SNR analysis, where the signal-to-noise ratios were compared between the frontal mask shown in Figure S3 and an anatomical mask comprising visual cortex (which typically has a good coverage during the data acquisition). This analysis, shown in Figure S8, did not indicate a significant difference between the frontal and visual areas (for details of this analysis see the Supplementary text, under Quality of signal in the frontal area).

2. For the DCM, is the overlap across modalities in the OFC being excluded? How robust are the results if this is not considered?

This is an interesting question. In our DCMs we extracted the activations based on univariate results shown in Figure 3 A-F, i.e. the red and blue activations corresponding to the auditory and visual values, respectively. We did not exclude the overlap between these activations as exploring the effective connectivity in subregions of modality-specific representations is beyond the scope of the current manuscript. The fact that we found evidence for modality-specificity, despite including overlapping regions of modality-specific OFC regions attests to the robustness of the modality-specific communication of valuation across the brain.

* The reviewers' comments are in black, our responses are in blue, and the revised sections in the main text are in green.

Point-by-point response

Reviewer #1 (Remarks to the Author):

The authors have done an excellent job addressing all three sets of reviewer comments.

Thank you for your positive evaluation of our work and for the insightful and constructive suggestions. These have greatly contributed to enhancing our analyses and interpretation of the results. We have incorporated your additional feedback into our work, as detailed below.

I have two small notes:

1) It is worthwhile to report post-hoc comparisons from the ANOVA analysis reported in Fig 3G, since the interaction effect may suggest that only one of the two regions represents modality specificity. Please include the direct comparisons between SV_vis and SV_aud in the visual ROI, and respectively - SV_vis and SV_aud in the auditory ROI (OFC).

Thank you for this suggestion. We have now incorporated it into lines 358-372 of the Results section, which is copied below for your reference (revised text is highlighted in green):

“Modality-specific valuation would require that the auditory ROI is more responsive to variations in the subjective value (SV) of auditory stimuli compared to visual stimuli, and vice versa for the visual ROI. Statistically, this would manifest as an interaction effect between ROI and the type of SV that the ROI represents. Conversely, modality-general valuation would require that auditory and visual ROIs respond similarly to both auditory and visual SVs. Examining auditory and visual ROIs in the OFC (**Figures 3G**, upper panel) and vmPFC (**Figures 3G**, lower panel) provided evidence for these types of valuation, respectively. A repeated measures ANOVA with factors: ROI (visual or auditory) in the OFC, and SV (visual or auditory), showed only an interaction between ROI and SV modality ($F[1,19] = 9.88$, $p = 0.005$), with no main effect of ROI or SV (all p s > 0.1). The interaction effect corresponded to numerically stronger responses of the auditory ROI to auditory SVs (mean \pm s.e.m: 0.57 ± 0.07) than to visual SVs (0.47 ± 0.01 , Cohen's $D = 0.28$), and similarly stronger responses of the visual ROI to visual SVs (mean \pm s.e.m: 0.60 ± 0.06) than to auditory SVs (0.50 ± 0.06 , Cohen's $D = 0.28$). However, in both cases, the differences between responses to the same versus different modalities did not reach statistical significance (p s > 0.1).”

2) I Deeply appreciate the new MVPA analysis, and think that it improves the paper significantly. However, I am not sure how to interpret the Y-axis in Figure 4. What is the chance level? Does the axis represent the change in accuracy level or overall accuracy?

Many thanks for your thoughtful comment. The Y-axis represents overall classification accuracy compared to the chance level. In all panels of Figure 4, we subtracted the accuracy of classifiers trained with shuffled class labels from the accuracy of each classifier. This approach allows us to present classification accuracy as the difference from the chance level. We have updated the legend of Figure 4 to clarify this, as shown below:

“Figure 4. Multivariate pattern analysis (MVPA) results. (A-B) MVPA results in the anatomical ROIs, defined based on the AAL atlas. Each boxplot represents the average classification accuracies for classifying the data into four levels of subjective value across all classifiers of the same type against the chance level (i.e., the accuracy of classifiers trained with shuffled labels was subtracted from each classifier's accuracy).”

This is a nice and interesting paper that will be of interest to any scholar interested in decision-making neuroscience.

Thank you again for your comprehensive and helpful suggestions which have substantially improved the quality of our work.

Reviewer #2 (Remarks to the Author):

The authors have done a good job responding to reviewers' comments. I do not have any further comments.

Reviewer #3 (Remarks to the Author):

The authors have addressed most of my concerns, and the paper has become much clearer and the results more convincing.

Thank you for your positive evaluation, your many insightful and constructive suggestions in the first round of revisions, and for clarifying the points below.

Comment part 1 - I am still having trouble understanding some of the decisions made to model the control task and conceptually what the modelled subjective value means in the control task. I originally thought this confusion was less crucial since it did not seem like the control task SV was used to model the neural signal but in their rebuttal the authors clarify that “we modelled these regressors across the entire experiment, including both value and control tasks within the same GLM”. ... These details aside, from a conceptual standpoint, if inferred SV in the control task is not relevant for behaviour it seems strange to include it in the neural model. I would appreciate clarity on anything I may have misinterpreted here.

Response to part 1 -

Thank you for your careful observations. It seems there may have been a misunderstanding due to a lack of clarity on our part.

For the GLM, the parametric regressors used in the control task **were not SV regressors and did not come** from the LNP model – since, as the Reviewer highlights, there were no meaningful SV representations in the control task. Instead, we modelled the control task regressors based on whether participants followed the instructions or not and which of the two options had been instructed. Our reasoning was that in the control task, the goal was to passively follow instruction (either to keep or switch the option in the next trial) as indicated by the feedback color of the preceding trial, which is captured by how we modelled the parametric regressors as shown below in Supplementary Table S4.

Additionally, it's important to note that while GLMs were constructed using data from all tasks and sessions, there was a regressor for task type. This design ensures that during periods when a specific task was being performed, only the corresponding task-specific regressors (e.g., parametric regressors for SV in the value task) were used to estimate neural responses. Consequently, the modeling approach for one task did not influence the neural response estimates for the other task.

For reference, we have copied below the definition of parametric regressors from the Methods section (lines 1000-1007) and the Supplementary Table illustrating the parametric regressors used in the control task:

“Similarly, for the control task we defined individually for each of the three modality-domains one unmodulated regressor representing the modality-wise trial identity and two parametrically modulated regressors corresponding to each of the options presented. In the control task, the aim was to passively follow instructions. Thus, to create a parametrically modulated regressor corresponding to one stimulus option, a weight of either 1 or 0 was assigned at the onset of stimuli options in each trial depending on whether the instruction (keep/switch your choice) from the last trial was correctly followed or not, respectively (see also Supplementary Table S4 and Figure S4).”

Table S4. Definition of parametric regressors for the control task*		
Choice on trial $t-1$	Instruction from trial $t-1$	Weight assigned on trial t
S_1	Not followed	$S_1 - 0, S_2 - 0$
S_2	Not followed	$S_1 - 0, S_2 - 0$
S_1	Followed; Instruction was to keep	$S_1 - 1, S_2 - 0$
S_1	Followed; Instruction was to switch	$S_1 - 0, S_2 - 1$
S_2	Followed; Instruction was to keep	$S_1 - 0, S_2 - 1$
S_2	Followed; Instruction was to switch	$S_1 - 1, S_2 - 0$

*When instruction from the previous trial ($t-1$) was not followed, a weight of 0 was assigned to both options S_1 and S_2 . When the instruction from the previous trial $t-1$ was correctly followed, the option that corresponded to the correct instructed choice in trial t received a weight of 1 and the other option received a weight of 0.

Comment part 2 - From clarifications made by the authors I now understand that the choices in the control task were modelled by considering the switch trials as analogous to rewarded trials. Which means they replace r (Equation 2) with the instructive feedback in the control task. This strikes me as odd since rewarded trials are more behaviourally similar to stay trials (i.e., both increase the probability of repeating the previous trial choice). If this is the case, not predicting the choice data well would be a result of mismodeling rather than an interpretable difference in learning in the two tasks.

Response to part 2

Thank you for your detailed comment. We agree that rewarded trials in the value task can be considered behaviourally more similar to "stay" trials in the control task, as both increase the probability of repeating the previous choice. However, "stay" trials are not fully analogous to the rewarded trials, as subjects often stayed with the same option even after not receiving reward, and conversely, occasionally switched to a different option after receiving reward. Most importantly, we demonstrate that adopting an alternative modeling approach that treats "stay" trials in the control task as analogous to rewarded trials in the value task yields similar results to those presented in our manuscript (see Reviewer Figure 1). We clarify this point in more detail below:

1. Effect on the LNP Model Outcome: In our behavioural analysis, "switch" trials in the control task were indeed modelled analogously to rewarded trials in the value task. But changing the convention to treat stay trials as rewarded does not affect the outcome of the LNP model. As described in lines 260-285, the LNP model uniquely explains choice behaviour in the value task but not in the control task. This difference is not due to a "mismodeling" of choices in the control task but is instead due to how participants adjust their beliefs based on feedback history in the two tasks. In the value task, participants perceive one feedback colour (the one which gives information on reward delivery) more favourable than the other. However, in the control task, there is no distinct desirability associated with the two feedback colours (both are equally "favourable" in order to perform well). The absolute differential value between the two options stays the same regardless of the feedback, because on each trial either one or another option takes on "high value" (correct response according to the instruction) and the other – "low value: (incorrect response). Reversing the feedback associations in the control task (i.e., yellow indicating "stay" and blue indicating "switch") and modelling "stay" trials analogously to rewarded

trials in equations 1-3 should not change the result of the LNP model. This is exactly what we found when we remodelled the data in the control task, using “stay” as analogous to reward in equations 1-3 (Reviewer Figure 1). Specifically, it still displays a flat slope in the control task, indicating that participants are not influenced by trial history beyond the instructions of the previous trial, differently to the value task. Since this approach does not change our conclusions or add any new insight, we did not include it in the manuscript. Additionally, the abovementioned points about the difference between the two tasks have been extensively elaborated in lines 260-285 of the Results section.

Reviewer Figure 1. The effect of different modelling approaches on the results of the control task. Please note that the results of the statistical comparison between the value and control tasks, as reported in lines 260-285, showed a consistent pattern across both modeling approaches—whether “switch” or “stay” trials were modeled similarly to “rewarded” trials. Specifically, when “stay” was modelled similarly to rewarded trials we saw:

- A significant interaction ($F[1,19] = 186.58, p < 0.001$) between the type of the task (value or control) and the type of feedback received in the past trial (blue or yellow).
- Whereas in the value task, absDVs derived from the model showed a significant difference between the two types of feedbacks (mean±s.e.m. = 0.12 ± 0.01 and 0.74 ± 0.03 for blue and yellow feedbacks, respectively, $p < 10^{-11}$); these estimates were not significantly different in the control task (mean±s.e.m. = 0.45 ± 0.01 and 0.48 ± 0.02 for blue and yellow feedbacks, respectively, $p = 0.0861$).

2. Purpose and Design of the Control Task: Conceptually, the main aim of the control task was to elicit similar sensory processing but distinct value processing mechanisms compared to the value task. To closely match the sensory processing requirements across both tasks, we used the same set of stimuli, trial structure (i.e., events within a trial), and importantly, a similar temporal choice pattern over the trials in a block. This was achieved by including a 33% switch rate in the control task, wherein participants passively switched their choices, on average, in one-third of the trials within a block when indicated by yellow feedback. Based on the Reviewer’s comment, we now confirmed that the average switch rate (across participants) in the value task was 27.12% (± 0.02), roughly similar to 33% switch rate in the control task. Thus, subjects also often stayed with the same option even after not receiving reward (which was only delivered in a third of trials) in the value task. This indicates that the design of the control task did capture the temporal structure of the value task quite well. This information is now added to the Methods section line 841-843 under the “Control Task Structure” section:

“To confirm that this design met its intended purpose, we examined participants’ choices in the value task and found an average switching rate of 27.12% ($\pm 0.02\%$ s.e.m.), closely aligning with the switch rate used in the control task.”